

# The 3D biogeochemical marine mercury cycling model MERCY v2.0 – linking atmospheric Hg to methyl mercury in fish.

Johannes Bieser[1,*], David Amptmeijer[1], Ute Daewel[1], Joachim Kuss[2], Anne L. Soerenson[3], Corinna Schrum[1,4],

[1] Helmholtz-Zentrum Hereon, Institute of Coastal Research, Max-Planck-Str. 1, 21502 Geesthacht, Germany
[2] Leibniz Institute for Baltic Sea research, Department for Marine Biogeochemistry, Seestraße 15, 18119 Rostock, Germany
[3] Swedish Museum of Natural History, Department of Environmental Research and Monitoring, Stockholm, Sweden
[4] Universität Hamburg, Institute for Marine Sciences, Mittelweg 177, 20146 Hamburg, Germany
[*] Correspondence to: Johannes Bieser (johannes.bieser@hereon.de)

## 10  Abstract

Mercury (Hg) is a pollutant of global concern. Due to anthropogenic emissions, the atmospheric and surface ocean Hg burden has increased substantially since preindustrial times. Hg emitted into the atmosphere gets transported on a global scale and ultimately reaches the oceans. There it is transformed into highly toxic methylmercury (MeHg) that effectively accumulates in the food web. The international community has recognized this serious threat to human health and in 2017 regulated
Hg use and emissions under the UN Minamata Convention. Currently, the first effectiveness evaluation of the Minamata Convention is being prepared and, in addition to observations, models play a major role in understanding environmental Hg pathways and in predicting the impact of policy decisions and external drivers (e.g. climate, emission, and land-use change) on Hg pollution. Yet, the available model capabilities are mainly limited to atmospheric models covering the Hg cycle from emission to deposition. With the presented model MERCY v2.0 we want to contribute to the currently ongoing effort to im-
prove our understanding of Hg and MeHg transport, transformation, and bioaccumulation in the marine environment with the ultimate goal of linking anthropogenic Hg releases to MeHg in sea food.

Here, we present the  equations and parameters implemented in the MERCY model and evaluate the model performance for two European shelf seas, the North-and Baltic Sea. With the model evaluation we want to establish a set of general quality criteria that can be used for evaluation of marine Hg models. The evaluation is based on statistical criteria developed for the
performance evaluation of atmospheric chemistry transport models. We show that the MERCY model can reproduce observed average concentrations of individual Hg species in water (normalized mean bias: $Hg_T$ 17%, $Hg^0$ 2%, MeHg -28%) in the two regions mentioned above. Moreover, it is able to reproduce the observed seasonality and spatial patterns. We find that the model error for $Hg_{T(aq)}$ is mainly driven by the limitations of the physical model setup in the coastal zone and the availability of data on Hg loads in major rivers. In addition, the model error in calculating vertical mixing and stratification contrib-
utes to the total $Hg_T$ model error. For the vertical transport we find that the widely used particle partitioning coefficient for organic matter of $\log(k_d)=5.6$ is too low for the coastal systems. For $Hg^0$ the model performance is at a level where further model improvements will be difficult to achieve. For MeHg, our understanding of the processes controlling methylation and demethylation is still quite limited. While the model can reproduce average MeHg concentrations, this lack in understanding hampers our ability to reproduce the observed value range. Finally, we evaluate Hg and MeHg concentrations in biota and
show, that modelled values are within the range of observed levels of accumulation in phytoplankton, zooplankton, and fish. The model performance demonstrates the feasibility of developing marine Hg models with similar predictive capability as established atmospheric chemistry transport models. Our findings also highlight important knowledge gaps in the dynamics controlling methylation and bioaccumulation that, if closed, could lead to important improvements of the model performance.





# 1 Background

Mercury (Hg) is a global pollutant and a dangerous neurotoxin (AMAP/UNEP, 2019). Since pre-industrial times, the global Hg cycle has been significantly altered by anthropogenic emissions (Streets et al., 2019) resulting in a three-fold pre-anthropogenic to present-day increase in the atmospheric and substantial increase in oceanic Hg burden (Lehnherr et al. 2015, Amos et al., 2013). The major anthropogenic sources of Hg are emissions from coal-fired power plants, small scale artisanal gold mining, and metal and cement production (Pirrone et al., 2010; AMAP/UNEP 2013, 2019). In addition, natural emissions and legacy reemissions from previously deposited Hg (most of it of anthropogenic origin) also contribute significantly to the atmospheric Hg burden (Pirrone et al., 2010; Driscoll, 2013; Obrist, 2018). The atmospheric lifetime of Hg is estimated in the range of 0.6 and 1.0 years (Slemr et al., 2018) resulting in a global atmospheric distribution of Hg. Atmospheric Hg will eventually deposit (Cohen et al., 2016; Jiskra et al., 2018). A large fraction is deposited directly to the ocean but Hg deposited to land can also be transported to the ocean via rivers and groundwater. In the aqueous phase, inorganic Hg can be methylated forming the highly bioaccumulative monomethylmercury (MMHg) and/or dimethylmercury (DMHg). These MeHg compounds are readily accumulated in the food web and pose a risk to food safety and human health (Clarkson, 1990; Mason et al., 1996; Chen et al., 2012; Parks et al., 2013; Puty et al., 2019). Because of this, the international community, under the umbrella of the United Nations Environmental Program (UNEP), signed the Minamata Convention on Mercury which came into force in 2017. Under this convention, all participating 184 nations have agreed to assess Hg pollution under their jurisdiction, to minimize usage and release of Hg compounds into the environment, and to regularly assess the impact of the reduction measures taken on environmental Hg burden and distribution. In order to assess the impact of reduction measures, there is an urgent need to understand the Hg pathways from anthropogenic releases to top predators and humans, with a specific attention to the marine ecosystem.

In this manuscript, we (1) introduce a newly developed numerical multi-compartment model for Hg cycling in the marine environment including accumulation in the marine food-web (MERCY v2.0) and (2) evaluate the model performance to reproduce observed concentrations, seasonality, and variability of Hg species. For the latter, we apply performance criteria used for evaluation of atmospheric chemistry transport models also for evaluation of marine Hg models. We use these criteria to (2.1) quantify the models predictive capabilities based on our current understanding of Hg cycling, to (2.2) identify the major sources of model error, and to (2.3) quantify the constraints on model improvement based on current process understanding and measurement availability and uncertainty. With this study, we present an evaluation of our marine Hg model and a general framework that provide the basis for future intercomparison studies of marine Hg models.



## 1.1 Research question

The key question concerning Hg pollution is how changing Hg emissions and other external stressors such as climate and land-use change impact MeHg accumulation in sea food which is an important global protein source for human consumption (Pauly et al., 2002; Obrist et al., 2018). To anticipate the natural Hg cycle and to identify the impact of human actions on the system it is necessary to develop multi-compartment chemistry transport models (CTMs) including all relevant compartments: atmosphere, soil/vegetation, rivers and oceans, sediments, and the marine ecosystem. The need to incorporate all compartments into a single multi-compartment model arises from the fact that Hg is non-degradable and constantly cycling between environmental compartments, unlike most pollutants which tend to accumulate in a single compartment and/or degrade over time. For example, atmospheric deposition of oxidized Hg is a major flux of Hg into the ocean but reduction reactions in the ocean and the high vapour pressure of elemental $Hg^0$ also result in a constant release of Hg from ocean to atmosphere (Fitzgerald et al., 1984; Kim and Fitzgerald, 1986; Andersson et al., 2008). The only real sink for Hg in the environment is burial in the lithosphere mainly as stable cinnabar (HgS) in anoxic marine sediments. Thus, coupled earth system models are needed to gain a deeper understanding of the processes and dynamics governing transport of Hg, Hg methylation and the variability of Hg accumulation in the marine food web. While there is a large number of emissions and atmospheric CTMs, there are still only a limited number of CTMs with a focus on the marine Hg cycling and food web transfers.

## 1.2 Development and state-of-the-art in Hg modelling

Atmospheric Hg modelling is well established and a large variety of global (ECHMERIT: Jung et al., 2009; De Simone et al, 2014; GLEMOS: Travnikov and Ilyin, 2009; Travnikov et al., 2009; GEM-MACH: Dunfoord et al., 2012; Kos et al., 2013; Dastoor et al., 2015; GEOS-Chem: Holmes et al., 2010; Amos et al., 2012; Song et al., 2015) and regional (CMAQ: Bullock et al., 2008; Bash et al., 2010; Zhu et al., 2015, DEHM: Chritensen et al., 2004; WRF-Chem: Gencarelli et al., 2017) atmospheric CTMs for Hg cycling have been published. Due to this abundance, many model inter-comparison and source apportionment studies have improved our understanding of atmospheric Hg transport, source receptor relationships and allowed us to predict future atmospheric Hg levels and deposition fluxes (Bergan et al., 1999; Xu et al., 2000; Petersen et al., 2001; Lee et al., 2001; Seigneur et al., 2001; Bullock et al., 2002; Dastoor et al., 2002; Hedgeock et al., 2004; Selin et al., 2007; Travnikov et al., 2009; Bieser et al., 2014; Gencarelli et al., 2014; Dastoor et al., 2015; Song et al., 2015; Cohen et al., 2016; Travnikov et al., 2017; Bieser et al., 2017; Horowitz et al., 2017). These models and studies are a keystone in informing policy makers to support the implementation and effectiveness evaluation of the Minamata Convention (http://www.mercuryconvention.org).

Compared to Hg modelling in the atmosphere, marine Hg modelling is still in its infancy and only a limited number of models exist so far. The development of marine Hg models can be divided in to four phases. At first, the ocean was implemented as a boundary for atmospheric CTMs and nowadays most atmospheric CTMs implement some kind of surface





ocean parametrization to explicitly include Hg air-sea exchange. One of the earliest marine Hg model developments was the addition of inorganic Hg red-ox chemistry and transport in a 2D slab ocean model coupled to the GEOS-Chem model (Selin et al., 2008; Strode et al., 2007; Soerensen et al., 2010). The aim of these early models was to improve air-sea exchange by including horizontal transport, red-dox chemistry, and river loads. Next, came the development of the first marine 3D models. These models, still limited to the inorganic Hg cycle, were used to investigate marine Hg dynamics (Zhang et al., 2014a; Zhang et al. 2014b; Bieser and Schrum 2016). In the next stage, several specialized marine Hg models were developed which were not based on 3D hydrodynamic models. Soerensen et al. (2016) published a coupled physical-biogeochemical multi-box model including organic Hg chemistry to investigate the Hg budgets in the Baltic Sea. Focusing on bioaccumulation Schartup et al. (2017) implemented Hg accumulation in a complex food web model and Sunderland et al. (2018) modelled the consumer exposure to MeHg in sea food. Finally, Pakhomova et al. (2018) developed a model with comprehensive Hg chemistry based on a hydrodynamic 1D model. Only in recent years has the development of comprehensive marine Hg models gained traction. So far, four marine Hg models based on numerical hydrodynamic 3D models have been published (Semeniuk et al., 2017; Zhang et al., 2019; Kawai et al., 2020; Rosati et al., 2022). All of these models include a complete marine Hg chemistry including MeHg. Yet, only Zhang et al. (2020) and Rosati et al. (2022) also implemented Hg cycling into a biogeochemical model considering uptake to and release from marine biota, making it the first hydrodynamic 3D Hg model to include the marine ecosystem.

## 1.3 Our contribution to the presented problem

Here we present our newly developed biogeochemical multi-compartment model for Hg cycling MERCY v2.0 and evaluate its predictive capabilities and limitations using evaluation criteria applied for performance evaluation of atmospheric CTMs (Derwent et al., 2010; Thunis et al., 2012; 2013; Carnevale et al., 2014). We focus on the implementation of the marine Hg cycle including a comprehensive marine Hg chemistry and partitioning scheme as well as bio-concentration and bio-magnification. We improve on the state-of-the-art by introducing an experimental upper trophic layer that simulate Hg and MeHg accumulation in fish. To our knowledge, MERCY v2.0 includes all currently known processes controling marine Hg cycling. The model is based purely on processes, reactions, and rates published in peer reviewed literature and no additional model tuning was performed.

We investigate the model predictive capabilities, something we consider important before using the model to study budgets or global dynamics. This allows us to quantify our model uncertainty, which for other models has only been loosely constrained to be *'orders of magnitude'* (Kawai et al., 2020), and discuss the processes and parameters driving it. Set up on a high-resolution regional domain covering a wide range of marine regimes in a region with high primary productivity and a relative abundance of observations we evaluate the ability to reproduce observed concentrations, seasonality, and variability of individual marine Hg species. Using common practice from atmospheric Hg modelling, we establish a quantitative benchmark for the capability of the model to reproduce actual observations of marine Hg concentration and speciation. Based





on this we discuss the major knowledge gaps and research questions that need to be tackled in order to improve our understanding of marine Hg cycling. Our ultimate goal is to improve capabilities to link changes in external stressors like anthropogenic emissions and climate change to MeHg accumulation in the marine food web by providing an independent model for marine Hg cycling and by fostering collaboration in the form of model inter-comparison studies comparable to the efforts in atmospheric Hg modelling (Ryaboshapko et al., 2002; Bullock et al., 2008; Travnikov et al., 2017; Bieser et al.,

2018). Finally, we want to identify and communicate the major needs for monitoring of Hg species in the marine environment.

## 2 Model description

### 2.1 Model Framework

The marine Hg chemistry scheme we develop for MERCY v2.0 is embedded in to GCOAST (Geesthacht Coupled cOAstal

model SysTem), a modelling framework coupling physical, chemical, and biological numerical models. It is an update and overhaul of MERCY v1.0 (Bieser and Schrum, 2016) which featured only inorganic Hg chemistry and no ecosystem interactions. As input, MERCY uses hourly model output from four types of 3D hydrodynamic models (atmospheric physics, atmospheric chemistry, marine physics and marine ecosystem) to drive the marine Hg speciation, transport, and bioaccumulation model. While this approach requires a large amount of storage capacity, it reduces the computational requirements and allows

the model to be easily run with input from alternative biogeophysical models. The external variables used by MERCY are listed in Table 1. In brief, models used in this work are:

**(1)** The regional weather and climate model **COSMO-CLM** (Rockel und Geyer, 2008) provides meteorological variables used to calculate air-sea exchange (temperature and wind speed) and photolytic reactions (surface short wave radiation).

**(2)** The atmospheric chemistry transport model **CMAQ-Hg** (Buyn and Schere, 2006; Zhu et al., 2015; Bieser et al., 2016) is

155 forced by COSMO-CLM meteorology and used to calculate atmospheric transport, chemistry, particle partitioning, and deposition for atmospheric trace gases. MERCY uses atmospheric Hg concentrations and deposition fluxes from CMAQ-Hg.

**(3)** The physical hydrodynamic ice-ocean model **HAMSOM** (Backhaus et al., 1984, Schrum and Backhaus, 1999). HAMSOM is directly coupled to the ecosystem model ECOSMO enabling it to represent the impact of the ecosystem on the hydrodynamics (e.g. light attenuation by biota). In MERCY the physical variables are used to calculate marine mercury trans-

160 port as well as temperature and salinity dependence of mercury cycling and speciation. The HAMSOM advection scheme is used to transport all Hg state variables.

**(4)** The marine en-to-end NPZD (Nutrient Phytoplankton Zooplankton Detritus) ecosystem model **ECOSMO** (Schrum et al., 2006, Daewel and Schrum, 2013, Daewel et al. 2019). ECOSMO is a 3D resolved food web model directly coupled with HAMSOM. It includes nutrients (nitrogen, phosphorus, and silica) and a food web based on a functional group approach with





3 phytoplankton species (diatoms, flagellates, and cyanobacteria), 2 zooplankton species (herbivore and omnivore), a mac-
165 robenthos and a pelagic fish group representing higher trophic levels. Additionally, oxygen, biogenic opal, detritus, and dis-
solved organic matter are considered, and the model includes a two-layer sediment compartment to simulate sedimentation
and resuspension. In MERCY detritus and dissolved organic matter determine the partitioning of Hg and MeHg and factors
such as light attenuation and oxygen concentration influence Hg speciation. Moreover, concentrations of the various species
of the model food web are used to calculate bioconcentration and biomagnification of Hg and MeHg.

| # | Name | Description | Unit | Source model |
|---|------|-------------|------|--------------|
| 1 | $T_a$ | Air temperature | °C | COSMO-CLM |
| 2 | $U_{10}$ | Wind speed at 10m | m s$^{-1}$ | COSMO-CLM |
| 3 | RSRF | Shortwave radiation at surface | W m$^{-2}$ | COSMO-CLM |
| 4 | GEM | Atmospheric Hg$^0$ concentration | ng m$^{-3}$ | CMAQ-Hg |
| 5 | GOM | Hg$^{2+}_{(g)}$ (GOM) deposition | kg ha$^{-1}$ | CMAQ-Hg |
| 6 | PBM | Hg$_P^{2+}_{(s)}$ (PBM) deposition | kg ha$^{-1}$ | CMAQ-Hg |
| 7 | $T_w$ | Water temperature | °C | HAMSOM |
| 8 | rho | Water pressure | Pa | HAMSOM |
| 9 | $U_w$ | Water U-velocity | m s$^{-1}$ | HAMSOM |
| 10 | $V_w$ | Water V-velocity | m s$^{-1}$ | HAMSOM |
| 11 | S | Salinity | PSU | HAMSOM |
| 12 | $dh_0$ | Surface layer elevation | m | HAMSOM |
| 13 | FLA | Flagellate biomass | mgC m$^{-3}$ | ECOSMO |
| 14 | DIA | Diatom biomass | mgC m$^{-3}$ | ECOSMO |
| 15 | CYA | Cyanobacteria biomass | mgC m$^{-3}$ | ECOSMO |
| 16 | ZOS | Herbivorous zooplankton biomass | mgC m$^{-3}$ | ECOSMO |
| 17 | ZOL | Omnivorous zooplankton biomass | mgC m$^{-3}$ | ECOSMO |
| 18 | FSH | Fish biomass | mgC m$^{-3}$ | ECOSMO |
| 19 | MAC | Macro benthos biomass | mgC m$^{-2}$ | ECOSMO |
| 20 | DOC | Dissolved organic carbon concentration | mgC m$^{-3}$ | ECOSMO |
| 21 | POC | Particulate organic carbon concentration | mgC m$^{-3}$ | ECOSMO |
| 22 | STOT | Sediment load | mgC m$^{-2}$ | ECOSMO |
| 23 | RTOT | Resuspended sediment | mgC m$^{-2}$ | ECOSMO |
| 24 | $O_2$ | Oxygen concentration | mgC m$^{-3}$ | ECOSMO |
| 25 | $SO_4^{2-}$ | Sulphate concentration | mgC m$^{-3}$ | ECOSMO |
| 26-28 | $P_x$ | 3 × Production rates for phytoplankton species (x) | mgC cm$^{-3}$ s$^{-1}$ * | ECOSMO |
| 29-46 | $F_{x,y}$ | 17 × Feeding rates for biological species (x) on species (y) | mgC cm$^{-3}$ s$^{-1}$ * | ECOSMO |
| 47-54 | $M_x$ | 7 × Mortality rates for biological species (x) | mgC cm$^{-3}$ s$^{-1}$ * | ECOSMO |

*Table 1: MERCY input variables and source models. (*rates for macro benthos are in mgC m$^{-2}$ s$^{-1}$)*



## 2.2 General Equations

MERCY v2.0 implements all processes we identified as relevant for marine (pelagic and benthic) Hg cycling into a 3D
ocean-ecosystem model. MERCY is based on basic principles describing Hg transport, transformation, and bioaccumulation.
It is set up on the same grid and domain as the coupled ocean ecosystem model ECOSMO-HAMSOM. Based on archived
hourly ECOSMO-HAMSOM output, it is effectively offline coupled to the marine hydrodynamic and ecosystem models. The
ECOSMO-HAMSOM model has been shown to accurately reproduce ecosystem dynamics in the coupled North Sea-Baltic
Sea system. The model equations and a model validation on the basis of nutrients are presented in detail by Daewel and
Schrum (2013), who showed that the model can reasonably simulate ecosystem productivity in the North Sea and the Baltic
Sea on seasonal up to decadal timescales. Using the same numerical approximations as described in Daewel (2019) the

change in concentration of Hg state variables over time $\frac{\delta C}{\delta t}$ is estimated by the prognostic equation (Eq. 1).

$$\frac{\delta C}{\delta t} = V \nabla C + w_d \frac{\delta C}{\delta z} + \frac{dz}{\delta z} \left( A_v \frac{\delta C}{\delta z} \right) + R_{(C,B)} \qquad \text{Eq. 1}$$

The physical transport terms for advection $V \nabla C$ with 3Dd velocity field $V = (u, v, w)$, vertical transport

$w_d \frac{\delta C}{\delta z}$ with sinking velocity $w_d$, and turbulent mixing $\frac{dz}{\delta z} (A_v \frac{\delta C}{\delta z})$ with diffusion coefficient $A_v$ and velo-

city $V$ are calculated by the hydrodynamic host model. At the upper and lower boundary of the water column, boundary
conditions are presented to account for air-sea exchange (Section 2.3.6) and sedimentation and resuspension (Section 2.3.5).
Each Hg state variable $C$ is subject to additional transformations $R_{(C,B)}$ which include chemical transformations

$Rc_{(C)}$ (Section 2.3.1), partitioning $Rp_{(C)}$ (Section 2.3.2), and biological uptake $Rb_{(C,B)}$ by ecosystem group $B$
(Section 2.3.4) (Eq. 2). Marine biota areis implemented in the ecosystem model following a functional group approach fur-
ther described in Section 2.3.4. All transformations $R_{(C,B)}$ are mass conserving transfer reactions which means that be-
sides emission inputs and inflow/outflow at the domain boundaries no Hg is added or removed from the system. The exact
formulation of $R_{(C,B)}$ differs for each Hg species in the model. In this section, we give a general overview of all possible
transformations while the exact formulae and parametrizations are given in Section 2.3. A complete list of all Hg state vari-
ables is given in Table 2. All chemical reactions $Rc_{(C)}$ and their respective reaction rates can be found in Table 3 and fur-
ther physical and biological parameters are given in Table (4).

$$R_{(C,B)} = Rc_{(C)} + Rp_{(C)} + Rb_{(C,B)} \qquad \text{Eq. 2}$$





Chemical transformations $\mathrm{Rc}_{(C)}$ (Eq. 3) are the sum of all reactions where species $C$ is a reaction product $\sum_{i=0}^{n} k_i C_i$ of another species $C_i$ with reaction rate $k_i$ minus the sum of all reactions where $C$ is an educt

$\sum_{j=0}^{n} k_j C$ with reaction rate $k_j$. Chemical reactions are implemented as pseudo 1$^{\text{st}}$ order reactions $\frac{\delta C}{\delta t}=kC$ either using a fixed reaction rate $k_1$ or a dynamic reaction rate $k_2=k_1 C_2$ dependent on a second reactant $C_2$ or an associated environmental variable (e.g. temperature). For photolytic reactions the reaction rate is $k=k_p E_\lambda$ with the integrated photon flux $E_\lambda=\int_{\lambda_0}^{\lambda_n} E_\lambda$ for specific wavelengths $\lambda$ and the photolysis rate $k_p$.

$$\mathrm{Rc}_{(C)}=\sum_{i=0}^{n} k_i C_i - \sum_{j=0}^{n} k_j C \qquad \text{Eq. 3}$$

$n$ = number of Hg species

Partitioning $\mathrm{Rp}_{(C)}$ (Eqs. 4) describes sorption and desorption of dissolved $C_{aq}$ to particulate organic matter $POM$ and dissolved organic matter $DOM$ where $C_{POM}$ is particulate Hg$^{2+}$$_{(s)}$ and $C_{DOM}$ Hg$^{2+}$$_{(aq)}$ bound to $DOM$. The equilibrium between these species is described by sorption $k_s$, $k_s'$ and desorption rates $k_d$, $k_d'$.

$$Rp_{(C)}(C_{aq})=k_d C_{POM}+k_d' C_{DOM}- k_s C_{aq} F \qquad \text{Eq. 4.1}$$

$$Rp_{(C)}(C_{POM})=k_s C_{aq} POM - k_d C_{POM} \qquad \text{Eq. 4.2}$$

$$Rp_{(C)}(C_{DOM})=k_s' C_{aq} DOM - k_d' C_{DOM} \qquad \text{Eq. 4.3}$$

Biological uptake $\mathrm{Rb}_{(C,B)}$ (Eq. 5) includes two distinct processes: (1) bio-concentration, which is defined as the passive uptake of dissolved Hg$^{2+}$$_{(aq)}$ through the cell membrane of a functional ecosystem group $B$ and (2) bio-magnification, which is the sum of active uptake and release through feeding. For higher trophic levels, the Hg in biota from active and pas-

sive uptake is stored in separate state variables with different release rates due to the differing accumulation patterns for each uptake process.





$$\mathrm{Rb}_{(C,B)} = \sum_{i=0}^{n} \left( v_i A_B C_i \right) + \sum_{b=0}^{m} \left( r_{B,b} \epsilon_C C_b - r_{b,B} C_B \right) - \left( r_{r(B)} + r_{m(B)} \right) C_B \qquad \text{Eq. 5}$$

$$n = \text{number of Hg species} \quad m = \text{number of ecosystem groups}$$

Bioconcentration $\sum_{i=0}^{n} \left( v_i A_b C_i \right) - r_r C_b$ is the sum of passive uptake with an uptake rate $r_u = v_i A_b$ depending on the

permeation velocity $v_i$ of dissolved Hg species $C_i$ and the average ecosystem group surface area $A_b$ minus an

ecosystem group and Hg species-dependent release rate $r_r$ multiplied with the Hg concentration inside biota $C_b$ .

Biomagnification $\sum_{b=0}^{m} \check{r}_{B,b} \epsilon_C C_b - \hat{r}_{b,B} C_B$ describes the active transfer of Hg driven by feeding rates $r_u = v_i A_b$ of an

ecosystem group $B$ on other ecosystem groups $b$ and the corresponding feeding pressure $\hat{r}_{b,B}$ . The efficiency of Hg

transfer upon feeding is determined by a Hg species-dependent uptake efficiency $\epsilon_C$ .

Additional release from the biological matrix $r_m C_b$ is described by a mortality rate $r_m$ . For the release of Hg from

detritus into the dissolved Hg pool $r_m$ is a temperature-dependent remineralization rate $\epsilon_D$ (see Eq. 9 in Section 2.3.1).

Finally, the respective change of dissolved Hg concentrations $\mathrm{Rb}_{(C,B)}$ due to uptake into and release from marine biota is

given by Eq. 6, where $\sum_{b=0}^{m} r_{B,b} (1 - \epsilon_C) C_b$ is the Hg fraction directly excreted into the dissolved phase upon feeding of

ecosystem group $B$ on another ecosystem group $b$ and $\left( r_{r(B)} + r_{m(B)} \right) C_B$ is the release due to a constant release

rate $r_{r(B)}$ and the mortality rate $r_{m(B)}$ of Hg species $C_B$ in ecosystem group $B$ .

$$\mathrm{Rb}_{(C,B)} = \sum_{b=0}^{m} \left\{ r_{B,b} (1 - \epsilon) C_b \right\} - \sum_{i=0}^{n} \left\{ v_i A_B C_i \right\} + \left( r_{r(B)} + r_{m(B)} \right) C_B \qquad \text{Eq. 6}$$

$$n = \text{number of Hg species,} \quad m = \text{number of ecosystem groups}$$



## 2.3 Implemented Processes

MERCY implements Hg using 35 variables (Table 2) representing different Hg species in the atmosphere, ocean, and sediment. For each model time step and each grid cell, the species are redistributed accounting for mass conservation based on physical, chemical, and biological processes. Figure 1 gives a graphical overview of transformations between Hg species in MERCY.

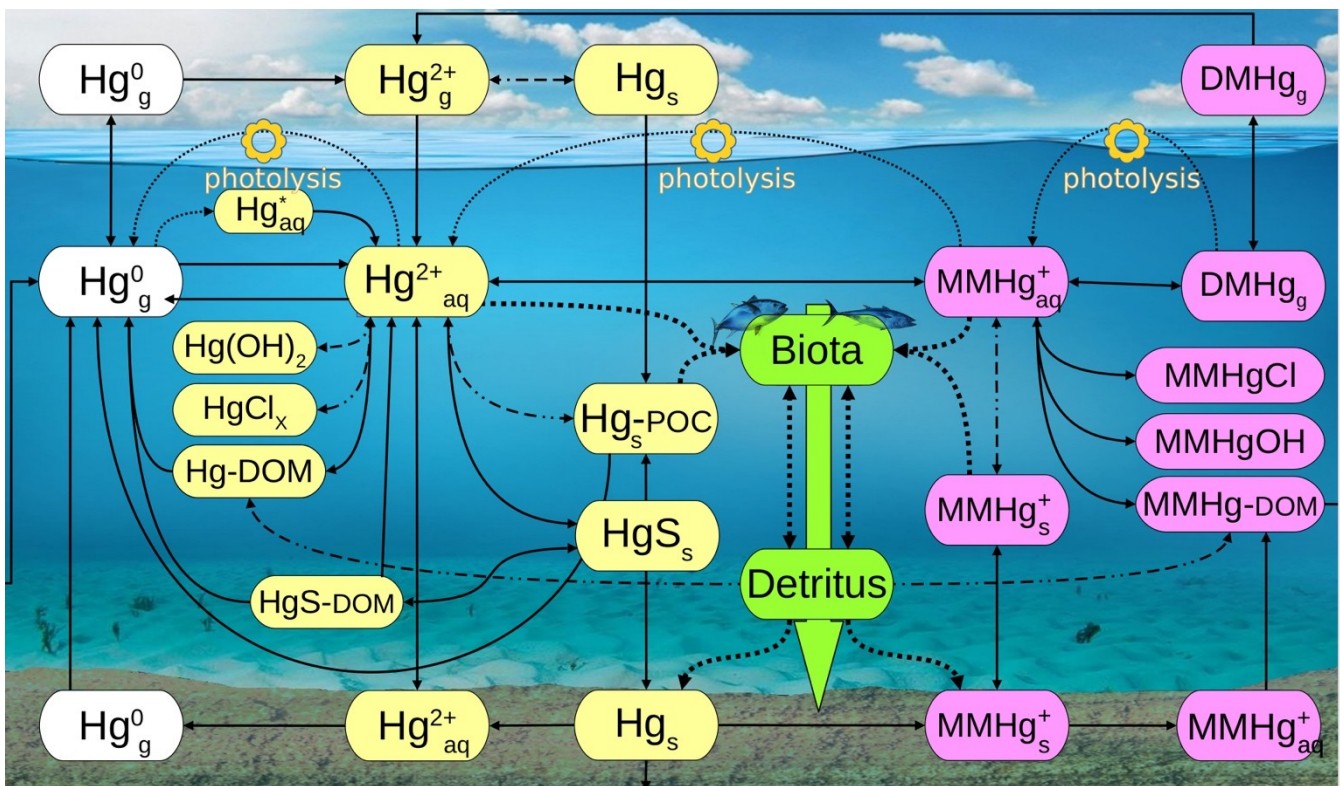

*Figure 1: Schematic of the chemical mechanism in MERCY. Solid lines indicate chemical reactions, fine dotted lines photolytic reactions, dahs-dotted lines instantaneous partitioning processes, bold dotted lines bioaccumulation and releases from biota into the dissolved phase. Colours codes are white for elemental mercury, yellow for inorganic oxidized mercury, pink for methylated mercury, and green for Hg in biota. The physical state of each species is indicated by: g for gaseous, aq for dissolved, and s for solid. The upper row indicates Hg species in the atmosphere, and the lower row those in the sediment.*

*All species and their reactions are given in Tabel 2 & 3.*





| Nr. | Species | Description | State | Compartments |
|---|---|---|---|---|
| 1-2 | $Hg^0_{(g)}$ | gaseous elemental mercury | gaseous | atmosphere, water |
| 3 | $Hg^{2+}_{(g)}$ | gaseous oxidized mercury | gaseous | atmosphere |
| 4-6 | $Hg_{(s)}$ | mercury bound to particulate matter | solid | atmosphere, water, sediment |
| 7 | $Hg\text{-}Det_{(s)}$ | mercury bound in detritus | solid | water |
| 8-14 | $Hg^{2+}_{(s)}$ | dissolved oxidized mercury accumulated inside biota | solid | biota* (see Section 2.3.4) |
| 15-18 | $Hg^{2+}_{(s)}$ | dissolved oxidized mercury attached onto biota | solid | biota* (see Section 2.3.4) |
| 19 | $Hg(OH)_{2(aq)}$ | mercury hydroxide | dissolved | water |
| 20 | $Hg(Cl)_{2(aq)}$ | mercury chloride | dissolved | water |
| 21 | $Hg\text{-}DOM_{(aq)}$ | mercury bound to dissolved organic matter | dissolved | water |
| 22 | $HgS_{(s)}$ | cinnabar | solid | water |
| 23 | $HgS\text{-}DOM$ | cinnabar bound to dissolved organic matter | dissolved | water |
| 24 | $MMHg^+_{(s)}$ | methyl mercury bound to particulate organic matter | solid | water |
| 25 | $MMHg\text{-}Det_{(s)}$ | methyl mercury bound in detritus | solid | water |
| 26-33 | $MMHg^+_{(aq)}$ | dissolved methyl mercury accumulated inside biota | solid | biota* (see Section 2.3.4) |
| 34-37 | $MMHg^+_{(aq)}$ | dissolved methyl mercury attached onto biota | solid | biota* (see Section 2.3.4) |
| 38 | $MMHgOH_{(aq)}$ | methyl mercury hydroxide | dissolved | water |
| 39 | $MMHgCl_{(aq)}$ | methyl mercury chloride | dissolved | water |
| 40 | $MMHg\text{-}DOM_{(aq)}$ | methyl mercury bound to dissolved organic matter | dissolved | water |
| 41-42 | $DMHg_{(g)}$ | dimethyl mercury | gaseous | atmosphere, water |

*Table 2: Hg species considered in MERCY. Species can represent state variables in multiple models. *Hg species in biota*
*($Hg^{2+}_{(s)}$ and $MMHg^{2+}_{(s)}$) represent one state variable for each functional group in the ecosystem model ECOSMO (see Section*
*2.3.4) giving a total of 42 species.*

### 2.3.1 Chemistry

In this section, we present all chemical state variables and the transformation processes in the model. A complete overview of all chemical transformations and the respective reaction rates $k$ is given in Table 3. All chemical transformations are calculated using pseudo-first-order reactions following Equation 7. The chemical mechanism is implemented using a tendency approach, where the relative change for each reaction is calculated and all changes to state variables are applied simultaneously. Equation 3 gives the change to the concentration of a single Hg species due to all reactions depleting and producing

it. We run the chemistry module with a time step of 60 s but find that it runs stable and efficient even with much larger time steps of 600 s.





$$C_t = C_0 \, e^{-t\,k}$$

Eq. 7

$C_t$ = Concentration at time = t      [ng/L]

$C_0$ = Concentration at time = 0      [ng/L]

t = time      [s]

k = pseudo-first order reaction constant      [$s^{-1}$]

| Nr. | Reaction | Description | Rate constant ($k_1$) | Source |
|---|---|---|---|---|
| R1 | $Hg^{2+} \rightarrow Hg^0$ | chemical (dark) reduction[1] | 3.00E-07 [$s^{-1}$] @ 0°C<br>6.00E-07 [$s^{-1}$] @ 15°C<br>7.50E-07 [$s^{-1}$] @ 20°C | Kuss et al., 2015 (see Eq. 8) |
| R2 | $Hg^{2+} + phot \rightarrow Hg^0$ | photolytic reduction | 1.00E-08 [$m^2\ W^{-1}\ s^{-1}$] | Kuss et al., 2015 |
| R3 | $Hg^{2+} \rightarrow Hg^0$ | biogenic reduction[2] | 8.06 E-09 [$m^3\ mg(C)^{-1}\ s^{-1}$] | Kuss et al., 2015 |
| R4 | $Hg^0 \rightarrow Hg^{2+}$ | chemical (dark) oxidation | 2.60E-06 [$s^{-1}$] | Kuss et al., 2015 |
| R5 | $Hg^0 + phot \rightarrow Hg^{2+}$ | photolytic oxidation[3] | 0.24E-08 [$m^2\ W^{-1}\ s^{-1}$] | Kuss et al., 2015 |
| R6 | $Hg^{2+} + H_2S \rightarrow HgS + 2H^+$ | cinnabar formation[4] | 4.90E-04 [$m^3\ mg(S^{2-})^{-1}\ s^{-1}$] | Slowey, 2010 |
| R7 | $HgS + DOM \rightarrow HgS\text{-}DOM$ | cinnabar dissolution | 5.78E-06 [$s^{-1}$] | Jiang, 2016 |
| R8 | $HgS\text{-}DOM_{(aq)} \rightarrow HgS_{(s)}$ | re-crystallisation | 9.50E-06 [$s^{-1}$] | Jiang, 2016 |
| R9 | $HgS + O_2 \rightarrow Hg^{2+} + 2SO_4^{2-}$ | cinnabar oxidation[5] | 1.00E-04 [$m^3\ mg(O_2)^{-1}\ s^{-1}$] | Petrochowa, 2019 |
| R10 | $Hg^{2+} \rightarrow MMHg^+$ | anoxic methylation | 4.40E-07 - 2.21E-07 [$s^{-1}$] | Monperuss et al., 2007 |
| R11 | $Hg^{2+} \rightarrow MMHg^+$ | constant methylation | 3.47E-08 [$s^{-1}$] | Duran et al., 2008 |
| R12 | $Hg^{2+} \rightarrow MMHg^+$ | biogenic methylation | 4.05E-09 [$l\ mg(C)^{-1}\ s^{-1}$] | Lehnherr et al., 2011<br>Olsen et al., 2018<br>Soerensen et al., 2018 |
| R13 | $Hg^{2+} \rightarrow DMHg$ | double methylation | 4.63E-10 [$s^{-1}$] | Lehnherr et al., 2011 |
| R14 | $MMHg^+ \rightarrow DMHg$ | bi-methylation | 1.51E-08 [$s^{-1}$] | Lehnherr et al., 2011 |
| R15 | $DMHg \rightarrow MMHg^+$ | de-methylation | 2.22E-09 [$s^{-1}$] | Mason, 1995; Mason, 1999 |
| R16 | $MMHg^+ \rightarrow Hg^{2+}$ | de-methylation | 6.94E-07 [$s^{-1}$] | Monperuss et al., 2007 |
| R17 | $DMHg + phot \rightarrow MMHg^+$ | photo de-methylation[3] | 4.57E-09 [$m^2\ W^{-1}\ s^{-1}$] | Lehnherr et al., 2007 |
| R18 | $DMHg + phot \rightarrow Hg^{2+}$ | photo de-methylation[3] | 4.57E-09 [$m^2\ W^{-1}\ s^{-1}$] | Lehnherr et al., 2007 |
| R19 | $MMHg^+ + phot \rightarrow Hg^{2+}$ | photo de-methylation[3] | 4.57E-09 [$m^2\ W^{-1}\ s^{-1}$] | Lehnherr et al., 2007 |
| R20 | $MMHg^+ \rightarrow Hg^0$ | reductive de-methylation | 2.22E-09 [$s^{-1}$] | Mason, 1995; Mason, 1999 |

*Table 3: Chemical reactions as implemented in the MERCY model. (pseudo $1^{st}$ order reaction rates* $k_2 = k_1 C$ *depend on*

*following variables C:* [1]*temperature dependent reaction rate,* [2]*cyanobacteria concentrationdependent reaction rate,* [3] *photo-*

*lytically active radiation dependent raction rate,* [4]*sulphate concentration dependent raction rate,* [5]*oxygen dependent con-*

*centration reaction rate.)*





### *Red-ox reactions*

Hg red-ox chemistry is implemented with five reactions. Reduction (Hg$^{2+}$ → Hg$^0$) is driven by three processes: (R1) a continuously ongoing chemical reduction, often referred to as dark reduction, (R2) photolytic reduction, and (R3) biogenic reduction (Table 3). We use reaction rates reported by Kuss et al. (2015). This leads to each reduction reaction roughly being of similar importance for the total Hg$^0$ production, albeit with specific distinct seasonality (note that the biogenic reduction only plays a role in the Baltic Sea due to cyanobacteria). This is in contrast to other published reaction rates where photolysis is

the dominant pathway (Qureshi et al., 2009). We do not use an intermediate oxidation product (Hg$^*$) as we found the species to be too short-lived for the given model setup. We chose the values from Kuss (2015) as contrary to other studies as these were measured under in situ conditions. The oxidation is driven mainly by chemical oxidation (R5) while photolytic oxidation (R6) rates are much smaller leading to a net photolytic reduction. The photolysis rates are parameterized to the photolytically active radiation based on observations. The biogenic reduction reaction rate is scaled by the cyanobacteria biomass and

is not triggered by other phytoplankton species (Kuss et al., 2015). For the chemical reduction, we consider a temperature-dependent reaction rate $k_{\mathrm{rd}}$ defined as 100% at 15°C (50% at 0°C and 125% at 20°C) (Kuss et al., 2015) (Eq. 8). Finally, we consider reductive demethylation of MeHg$^+$ (R19), which is only a minor source of Hg$^0$ in the model.

$$k_{\mathrm{rd}} = 2.92\text{E-}07\,e^{0.045\,T_w}$$  Eq. 8

$\quad T_w$ = Water temperature $\qquad$ [°C]

$\quad k_{\mathrm{rd}}$ = dark reduction rate $\qquad$ [s$^{-1}$] $\qquad$ (R1, Table 3)

### *Cinnabar formation*

Additionally, we implemented Hg sulphur chemistry using oxygen concentrations calculated by ECOSMO, whereas sulphur

ions (S$^{2-}$) are represented by negative oxygen concentrations in order to reduce the amount of transported state variables (Table 1). In anoxic waters cinnabar (HgS) is formed by reaction with sulphide species (H$_2$S, HS$^-$, S$^{2-}$) (R6, Table 3). This reaction is kinetically fast and scavenges the majority of the inorganic Hg$^{2+}_{(aq)}$ within a few hours. The product of this reaction is considered particulate but without a sinking velocity due to the small size of these particles (Paquette and Helz, 1995, Soerensen et al., 2018). In a slower reaction (R7) HgS is subsequently binding to -SH groups of DOM, a reaction that can lead

to the dissolution of 50% of the HgS within 24 hours. After one day, the dissolution reaction is in equilibrium with the recrystallisation reaction (R8). In the presence of oxygen, sulphur is quickly oxidized and HgS is readily transformed back into soluble Hg$^{2+}_{(aq)}$ species (HgS$_{(s)}$ + 2O$_2$ → HgSO$_{4\,(aq)}$) (R9). In the model, HgSO$_4$ is attributed back to the dissolved Hg$^{2+}_{(aq)}$ pool and not tracked by an additional state variable.




*Organic chemistrcy*

The organic chemistry doubles the number of variables introduced for the inorganic Hg chemistry mechanism (Figure 1). In the model, we implemented three sources for MMHg⁺, (1) methylation in anoxic waters (R10), (2) methylation in oxic waters (R11), and (3) methylation due to biologic activity (R12). The anoxic methylation is thought to be due to anaerobic bacteria

and is in our model the fastest methylation process ($4.4E\text{-}07$ $s^{-1}$). Studies have found that methylation also occurs in oxic waters although at much slower rates (Lehnherr, 2014; Heimbürger et al., 2015; Bowman et al., 2020; Soerensen et al. 2018). We implemented an additional constant methylation reaction ($3.47E\text{-}08$ $s^{-1}$) and a biologically induced methylation in oxic water to reflect the fact that numerous bacteria have been shown to actively methylate Hg (Soerensen et al., 2018; Capo et al., 2020). We use the amount of remineralized organic material as a proxy for anoxic micro environments in the oxic water

column. The remineralization is dependent on temperature (Eq. 9) with DOM being mineralized at a higher rate of

$k_{\text{rem}_{\text{DOM}}} = 10\, k_{\text{rem}_{\text{POM}}}$ . Following equation 7 we calculate the amount of remineralized organic matter and use this to scale the biologic methylation rate (R12). The reaction rate R12 has been chosen such that the effective biological methylation rate mostly lies between R10 and R11 ranging from zero to $1.13E\text{-}07$ $s^{-1}$ .


$$k_{\text{rem}_{\text{POM}}} = 0.006 \left\{ \left(1 + 20^{\left(\frac{T_{w^2}}{13^2 + T_{w^2}}\right)}\right)\right\} \qquad \text{Eq. 9}$$

$k_{\text{rem}_{POM}}$ = POC reminerlization rate  [day$^{-1}$]

$T_w$  = water temperature  [°C]

Besides MMHg⁺ we also consider double methylation reactions producing DMHg (R 13,14). For the degradation DMHg →

MMHg⁺ → Hg²⁺, we consider constant demethylation reactions (R15,16), photolytic degradation (R17-19), and reductive demethylation (R20). Finally, we apply methylation and de-methylation only to dissolved Hg²⁺(aq) and MeHg⁺(aq) species. Thus, high loads of DOM and POM influence the effective net methylation and produce a non-linear behaviour in the system (Olsen et al., 2018).

*Chemical reactions in the sediment*

In the sediments, we consider only two species: Hg²⁺(S) and MMHg⁺(s). These undergo methylation and demethylation using the same reactions and rates as in the pelagic zone (Table 3). We consider the sediments to always be at least partially anoxic depending on the oxygen concentration in the adjacent water layer (50 – 100% anoxic for O₂ between 2 and 0 ml/l). All abiotic methylation reactions (R10 and R11, Table 3) thus take place in the model sediment. Additionally, Hg²⁺(s) is subject to dark reduction and subsequently released from the sediment as Hg⁰ (Capo et al., 2022).






### 2.3.2 Partitioning

The speciation of $Hg^{2+}$ and $MMHg^+$ plays a major role in transport, chemical reactions, and bio-availability. In the partitioning scheme we distinguish between three phases: (1) dissolved $Hg^{2+}_{(aq)}$ and $MeHg^+_{(aq)}$ which are stored in two advected state variables. They are further resolved into $Hg(OH)_{2(aq)}$, $HgOHCl_{(aq)}$, $HgCl_{2(aq)}$, $MeHgOH_{(aq)}$, $MeHgCl_{(aq)}$ which are diagnostic
variables dependent on salinity. (2) Hg bound to dissolved organic material $Hg^{2+}$-$DOM_{(aq)}$ and $MeHg^+$-$DOM_{(aq)}$, and (3) the particulate Hg species $Hg^{2+}$-$POC_{(s)}$ and $MeHg^+$-$POC_{(s)}$.

Three-way partitioning is calculated as a function of Hg concentration, particle load, and dissolved organic matter concentration (Eqs. 10-13). As we could not obtain sorption and desorption rates and because our carbon representation does not capture the amount of O- and S- binding sites available for Hg we implemented partitioning based on partitioning coefficients
instead of a dynamic sorption/desorption process as described in Eqs. 4. We use a value of $\log(k_d) = 6.6$ for $Hg^{2+}$ associated with DOC based on the work of Tesan et al. (2020). This $K_d$ is higher than what is used in other models (Zhang et al., 2019; Kawai et al., 2020). Moreover, we use distinct partitioning coefficients for binding to POC $(k_d)$ and DOC $(k_l)$ for inorganic $Hg^{2+}$ ( $\log(k_d)$ = 6.4 and $R_{(C,B)}$ = 6.6) and organic $MMHg^+$ ( $R_{(C,B)}$ = 5.9 and $\log(k_l)$ = 6.0) (Allison and Allison, 2005; Batrakova et al., 2014) (Table 4).

$$k_d = \frac{Hg_{POC}}{POC * Hg^{2+}_{aq}} \quad \text{Eq. 10a} \qquad K_l = \frac{Hg_{DOC}}{DOC * Hg^{2+}_{aq}} \qquad \text{Eq. 10b}$$

$$Hg_{aq} = \frac{Hg_T}{1 + k_d + k_l} \quad \text{Eq. 11a} \qquad Hg_{DOC} = \frac{Hg_T k_d}{1 + k_d + k_l} \quad \text{Eq. 11b} \quad Hg_{POC} = \frac{Hg_T k_l}{1 + k_d + k_l} \quad \text{Eq. 11c}$$

$k_d$ = Hg-$POM_{(s)}$/$Hg_{(aq)}$ partitioning coefficient     []

$k_l$ = Hg-$DOM_{(aq)}$/$Hg_{(aq)}$ partitioning coefficient     []

POM = particulate organic matter     []

SPM = suspended particles     []

DOM = dissolved organic matter     []

SPM = suspended particles     []

DOM = dissolved organic matter     []


The model assumes instantaneous equilibrium and redistributes $Hg^{2+}$ and $MeHg^+$ between the three states on each time step. This approach is supported by lab studies that indicate the partitioning equilibrium is reached within an hour (Mason et al. 1994). Finally, mass conservation is ensured by Equation 12.

$$Hg_T = Hg_{aq} + Hg_{POC} + Hg_{DOC} \qquad \text{Eq. 12}$$





| # | Name | Description | Value | Unit | Source |
|---|------|-------------|-------|------|--------|
| 1 | R2P | Conversion factor short wave radiation to PAR | 0.5211 | unitless | Jacovides et al., 2004 |
| 2 | E2W | Conversion factor Einstein to Watt | 4.57 | [] | McCree et al., 1972 |
| 3 | PSR | Conversion factor POC to total particles | 0.1 | unitless | Sharif et al., 2014 |
| 4 | $F_{red}$ | Reducible fraction of dissolved $Hg^{2+}$ | 0.4 | unitless | Mason et al., 1995; Kuss et al., 2015 |
| 5 | $v_{down}$ | Particle settling velocity | 5.0 | m/s | Daewel and Schrum, 2013 |
| 6 | $v_{up}$ | Upwelling velocity of cyanobacteria | 0.1 | day$^{-1}$ | Daewel and Schrum, 2013 |
| 7 | $r_{bur}$ | Burial rate | 0.00001 | day$^{-1}$ | Daewel and Schrum, 2013 |
| 8 | $v_{crit}$ | Critical velocity triggering resuspention | 0.01 | m/s | Daewel and Schrum, 2013 |
| 9 | $r_{res}$ | Resuspention rate | 25 | day$^{-1}$ | Daewel and Schrum, 2013 |
| 10 | FRR | Remineralization fraction DOM/POC | 0.4 | unitless | Daewel and Schrum, 2013 |
| 11 | $p_{HgCl2}$ | Permeability of $HgCl_2$ | 7.2E06 | m/s | Mason et al., 1996 |
| 12 | $p_{CH3HgCl}$ | Permeability of $CH_3HgCl$ | 7.4E06 | m/s | Mason et al., 1996 |
| 13 | $\log(kd_0)$ | Partitioning coefficient of $Hg^{2+}$ | 6.4 | l/kg | Tesan et al., 2020 |
| 14 | $\log(kl_0)$ | Partitioning coefficient of $Hg^{2+}$ | 6.6 | l/kg | |
| 15 | $\log(kd_0)$ | Partitioning coefficient of $MeHg^+$ | 5.9 | l/kg | Allison and Allison, 2005 |
| 16 | $\log(kl_0)$ | Partitioning coefficient of $MeHg^+$ | 6.0 | l/kg | |
| 17 | $E_{H2O}$ | PAR extinction coefficient in water | 0.05 | m$^{-1}$ | Daewel and Schrum, 2013 |
| 18 | $E_{phy}$ | PAR extinction coefficient of phytoplankton | 3.77E-04 | m²/mgC | Daewel and Schrum, 2013 |
| 19 | $E_{DOC}$ | PAR extinction coefficient of DOC | 2.90E-04 | m²/mgC | Daewel and Schrum, 2013 |
| 20 | $E_{POC}$ | PAR extinction coefficient of POC | 2.0E-04 | m²/mgC | Daewel and Schrum, 2013 |

*Table 4: Physical and biological constants used in MERCY v2.0.*

### 2.3.3 Radiation

The radiation available for photolytic reactions is determined from hourly input fields using short wave radiation reaching the surface as modelled by the meteorological model COSMO-CLM (Table 2). As the reaction rates for Hg photolysis are usu-

ally reported in relation to photolytically active radiation $PAR$, we convert the modelled shortwave radiation using an average factor of 0.5211 not taking into account diurnal variations (Jacovides et al., 2004). We then calculate the cumulative light extinction $E_{tot}$ (Eq. 13) by water (Eq. 14), phytoplankton (Eq. 15), dissolved organic matter (Eq. 16), and suspended particles $SPM$ (Eq. 17). Whereby we estimate the total particulate matter concentration for light attenuation using a constant ratio of 0.1 times the particulate organic carbon $POC$ concentration (Sharif et al., 2014) (Eq. 18). Finally, the re-

maining radiation $R_z$ at half the depth of each layer is calculated following the Lambert-Beer Law (Eq. 19). All parameters used to calculate light extinction are given in Table 4.





$$E_{tot} = E_{phy} + E_{DOC} + E_{DOC} + E_{Water} \qquad \text{Eq. 13}$$

$$E_{H2O} = \sum_{z=0}^{n} 0.05 (h_{z+1} - h_z) \qquad \text{Eq. 14}$$

$$E_{phy} = \sum_{z=0}^{n} 0.000377 (C_{FLA} + C_{DIA} + C_{CYA}) \qquad \text{Eq. 15}$$

$$E_{DOC} = \sum_{z=0}^{n} 0.00029 \, C_{DOC} (h_{z+1} - h_z) \qquad \text{Eq. 16}$$

$$E_{POC} = \sum_{z=0}^{n} \frac{0.0002 \, C_{POC}}{PSR} (h_{z+1} - h_z) \qquad \text{Eq. 17}$$

$$C_{P_{total}} = C_{POC} / PSR \qquad \text{Eq. 18}$$

$$R_{z+1} = R_z \exp^{(-E_z)} \qquad \text{Eq. 19}$$

| | | |
|---|---|---|
| $C_{FLA}$ = Flagellate concentration | [mgC/m³] | |
| $C_{DIA}$ = Diatome concentration | [mgC/m³] | |
| $C_{CYA}$ = Cyanobacteria concentration | [mgC/m³] | |
| $C_{DOC}$ = Dissolved organic carbon | [mgC/m³] | |
| $C_{POC}$ = Particulate organic carbon | [mgC/m³] | |
| $C_{P\,total}$ = Total particle load | [mgC/m³] | |
| PSR = Fraction of POC to total particles | [] = 0.1 (Sharif et al., 2014) | |
| $E_{phy}$ = Extinction by phytoplankton | [] | |
| $E_{DOC}$ = Extinction by DOC | [] | |
| $E_{POC}$ = Extinction by POC | [] | |
| $E_{H2O}$ = Extinction by water | [] | |
| $E_{tot}$ = Total light extiction | [] | |
| z = number of vertical layer | [] | |
| n = number of layers | [] | |
| h = height of grid cell z | [m] | |
| R = Radiation at layer z+1 | [W/m²] | |





### 2.3.4 Biological uptake

Hg bioaccumulation has been implemented directly into the HAMSOM-ECOSMO framework (Daewel and Schrum, 2013; Daewel et al., 2019). ECOSMO is based on a functional group approach lumping species based on properties like nutrient requirements ($NO_3^-$, $NH_4^+$, $PO_4^{3+}$, $SiO_2$) and feeding habits (herbivorous, omnivorous, carnivorous). ECOSMO includes 3 phytoplankton species (flagellates, diatoms, and cyanobacteria), 2 zooplankton species (micro- and mesozooplankton), as well as a macrobenthos and a fish group with the latter representing mass fluxes to higher trophic levels (Figure 2).

In MERCY we consider bioaccumulation of inorganic $Hg^{2+}$ and organic $MeHg^+$, for each of the 7 functional groups. Moreover, we distinguish between passive uptake directly from the water column (bio-concentration) and active uptake due to the consumption of contaminated food (bio-magnification). The first is accumulated as Hg attached to the organism (zooplankton carapace, fish gills) and the second incorporated internally. Figure 3 depicts a schematic overview of the rate constants used to describe bioaccumulation in MERCY with phytoplankton, which only undergoes passive uptake, on

the left and higher trophic species, which also actively feed on other species, in the middle and on the right. All bioaccumulation processes are calculated separately for inorganic $Hg^{2+}$ and organic $MeHg^+$ and the accumulated Hg is transported consistently with the movement of the associated biota. In total, this leads to 22 bio-accumulation state variables (6 phytoplankton, 8 zooplankton, 4 macrobenthos, and 4 fish) which roughly doubles the number of chemical state variables (20) in the model (Table 2). All parameters used for bioaccumulation modelling are given in Table 5.

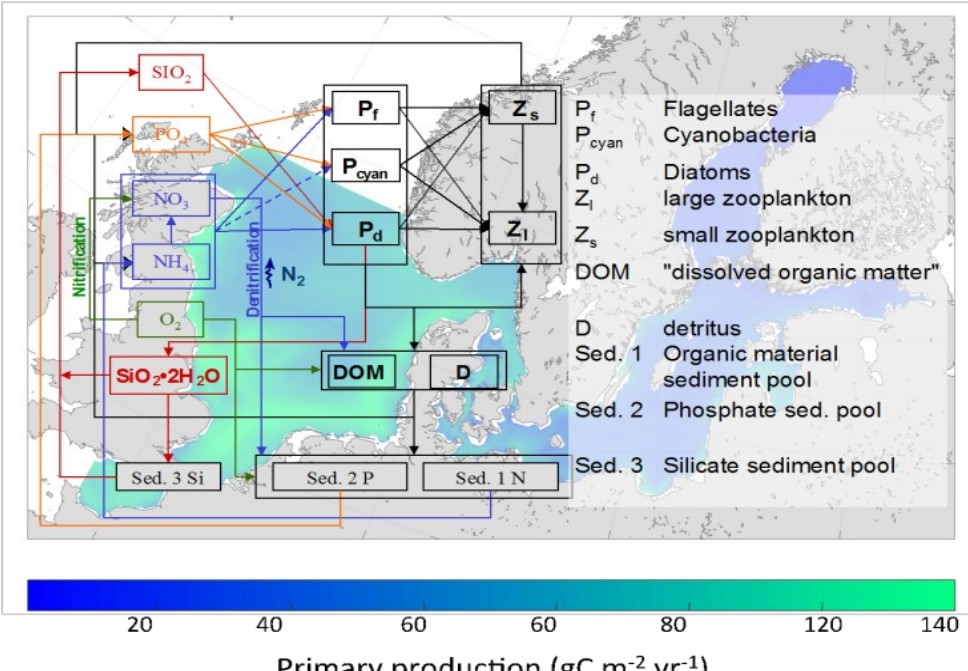

*Figure 2: Overview of the ECOSMO marine ecosystem nutrient and functional group model (Daewel et al., 2019).*



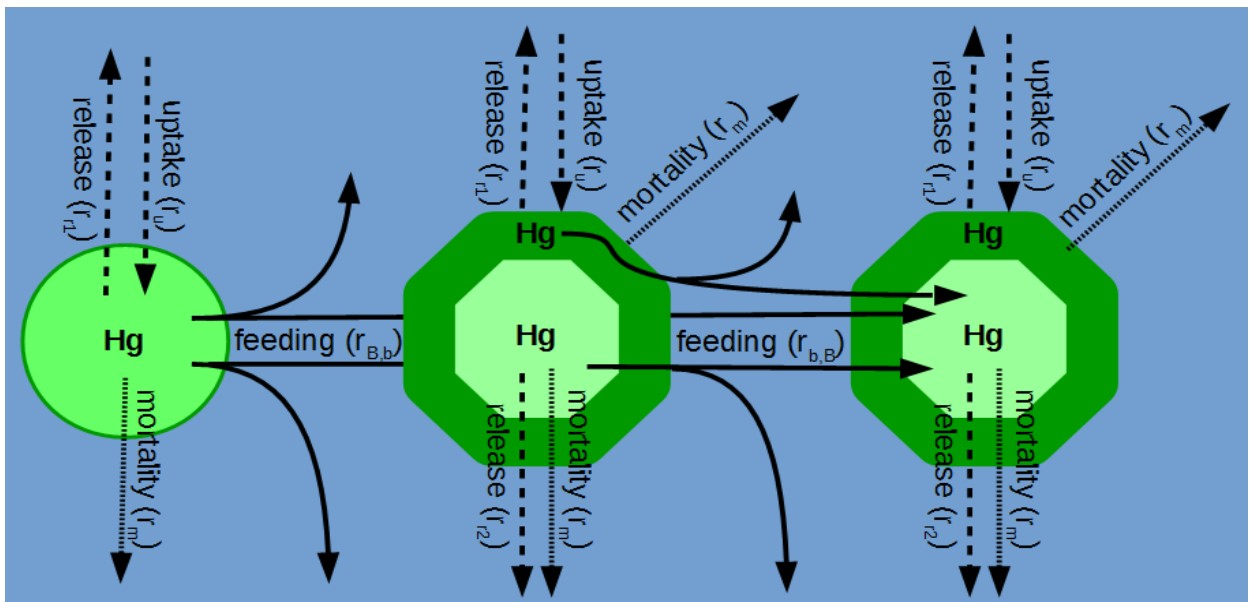

*Figure 3: Schematic overview of Hg$^{2+}$ and MMHg$^+$ bioaccumulation for phytoplankton (left), microzooplankton (middle), and mesozooplankton (right). Dashed lines indicate passive uptake and release rates (Eq. 19), solid lines indicate active up-take due to feeding with a fraction beeing instantly released back into the water column (Eq. 22), and dotted lines show Hg loss due to mortality (Eq. 23).*

### *Bio-concentration*

In MERCY dissolved Hg$^{2+}_{aq}$ and MMHg$^+_{aq}$ are accumulated via passive uptake $U_P$ (Eq. 20) through the cell membrane of the phytoplankton functional groups (diatoms, flagellates, cyanobacteria). For zooplankton, macrobenthos, and fish, the passive uptake is thought to lead to Hg accumulation on the surface or areas that are exposed to water like the mouth or gills in the case of fish (Figure 3: r$_u$). The uptake rate is $r_u$ calculated based on an ecosystem functional group $B$ dependent surface area $A_{(B)}$ and a Hg species dependent permeation velocity $v$ (Eq. 21). We estimate average volume and surface areas for phytoplankton species based on observations of size and geometric shape (Table S1) (Olenina et al., 2003). The cell volume is used to estimate the organic carbon content, which is then used to estimate the organic carbon to cell surface ratio (Menden-Deur and Lessard, 2000). This ratio allows us to model the total phytoplankton cell surface per functional group based on the organic carbon content as modelled in ECOSMO. The estimated surface area is used to calculate the Hg species-dependent uptake rate based on Mason et al. (1996). Diffusive uptake by zooplankton is implemented based on experimental uptake studies but is less important compared to phytoplankton due to the comparably low surface areas of these species (Tsui and Wang, 2004).





$$U_{P(B)} = r_u Hg^{2+}_{(aq)}$$ Eq. 20

$U_{P(B)}$ = passive uptake of ecosystem group B          [ng/s]

$r_u$ = passive uptake rate          [s⁻¹]

$Hg^{2+}_{(aq)}$ = dissolved Hg          [ng/m³]

$$r_u = v_C A_B C_B$$ Eq. 21

$v_i$ = permeation velocity for Hg species i          [m/s]

$A_B$ = average surface area of ecosystem group B          [m²/mgC]

$C_B$ = concentration of ecosystem group B          [mgC/m³]

*Bio-magnification*

For all non-phytoplankton species, we consider the active uptake $U_A$ due to feeding rates $r_{B,b}$ and $r_{b,B}$ which lead

to a fraction $\epsilon_{(C)}$ of the Hg in prey to be incorporated into the predator (Figure 3: r_{B,b} r_{b,B}). Through this process, Hg²⁺ and

MMHg⁺ are magnified along the food web (Eq. 22). Zooplankton is feeding on detritus, phytoplankton and other zooplankton

and finally consumed by fish (Figure 4). Moreover, there is macro benthos that exists only in the marine bottom layer and is

feeding on these species. We base our uptake on studies that show that only a fraction of Hg²⁺ ( $\epsilon_{(Hg)}$ = 0.45) and MMHg⁺

( $\epsilon_{(MeHg)}$ = 0.97) are incorporated into the predator, while the rest is excreted directly back into the water column (Mason

et al., 1996;, Wang and Wong, 2003;, Tsui and Wang, 2004; Pickhardt et al., 2006).

$$U_{A(B)} = \sum_{b=0}^{m} r_{(B,b)} \epsilon_{(C)} Hg_{(b)} - r_{(b,B)} Hg_{(B)}$$ Eq. 22

$U_{A(B)}$ = active uptake rate in ecosystem group B          [ng/s]

$Hg_B$ = Hg concentration in ecosystem group B          [ng/m³]

$Hg_b$ = Hg concentration in ecosystem group b          [ng/m³]

m = number of ecosystem groups          []

$r_{B,b}$ = feeding rate          [m³/s]

$r_{B,b}$ = predation rate          [m³/s]

$\epsilon_C$ = feeding efficiency          [dimensionless between 0-1]





***Release***

Mercury accumulated by active $U_P$ and passive uptake $U_A$ can also be released back into the water column (Eq. 23).
There are three distinct processes in the bioaccumulation model that release Hg accumulated in the food web back into the
water column. Firstly, there are species-dependent fixed release rates for Hg inside $r_{r2}$ and on $r_{r1}$ the biological spe-
cies (Eq. 24). Secondly, upon feeding described by feeding rates $r_{B,b}$ and $r_{b,B}$, a fraction $1-\epsilon_{(C)}$ of the Hg accu-
mulated in prey is not incorporated into the predator and this is directly released back into the water column (Eq. 25). Finally,
based on the ECOSMO mortality and respiration rates $r_m$ for each ecosystem group, Hg is released (Eq. 26). Feeding,
mortality and respiration rates are directly taken from ECOSMO (Table 1) and the relevant equations ar described in detail in
Daewel (2019). For detritus, the mortality rate is a temperature-dependent remineralization rate (Eq. 9).

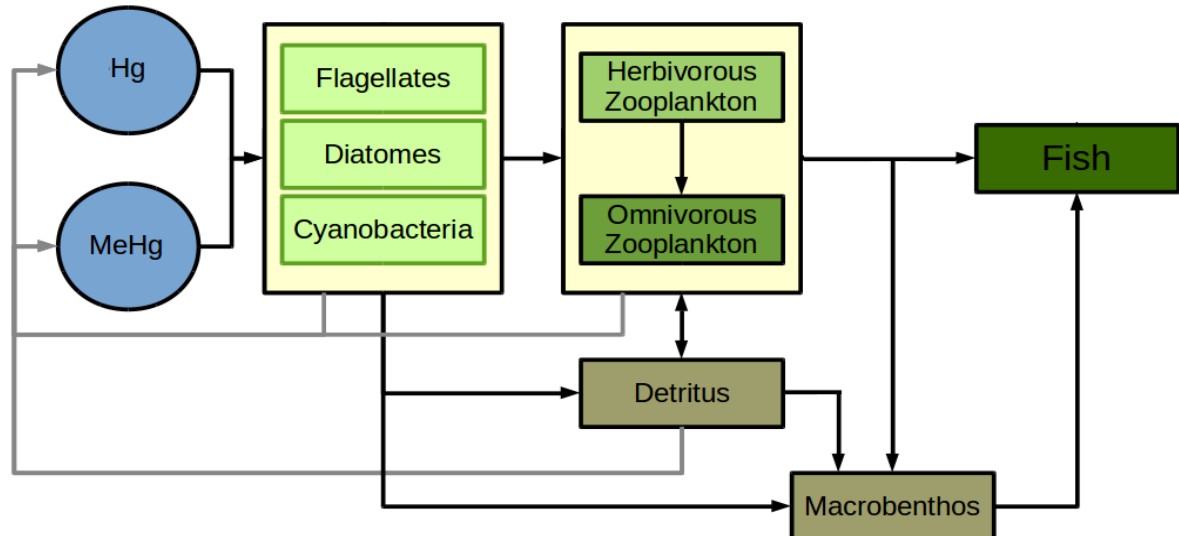

*Figure 4:  Flowchart of Hg bio-accumulation due to feeding following the ECOSMO end-to-end functional group approxim-
ation (Daewel, 2019). Rates for all depicted flows are given in Table 5.*


$$R_{(B)}=R_{F(B)}+R_{R(B)}+R_{M(B)}$$    Eq. 23

$R_{(B)}$ = release rate from ecosystem group B    [ng/s]

$R_{R\,(B)}$ = constant release rate    [ng/s]

$R_{F\,(B)}$ = feeding related release rate    [ng/s]

$R_{M\,(B)}$ = mortality related release rate    [ng/s]



$$R_{\mathrm{F}(B)}=\sum_{B=0}^{m}\{\sum_{b=0}^{m}r_{(B,b)}[(1-\epsilon_{(C)\mathrm{ext}})\mathrm{Hg}_{(b)\mathrm{ext}}+(1-\epsilon_{(C)\mathrm{int}})\mathrm{Hg}_{(b)\mathrm{int}}]\}\qquad\text{Eq. 24}$$

$r_{B,b}$ = feeding rate of group B on group b      [s⁻¹]

$\epsilon_{C\,\mathrm{int}}$ = feeding efficiency for external Hg species C      [dimensionless between 0-1]

$\epsilon_{C\,\mathrm{ext}}$ = feeding efficiency for external Hg species C      [dimensionless between 0-1]

$\mathrm{Hg}_{(b)\,\mathrm{ext}}$ = Hg on ecosystem group b      [ng/m³]

$\mathrm{Hg}_{(b)\,\mathrm{int}}$ = Hg inside ecosystem group b      [ng/m³]

$$R_{\mathrm{R}(B)}=\sum_{B=0}^{m}\square r_{\mathrm{r1}(B)}\mathrm{Hg}_{(B)\mathrm{ext}}+r_{\mathrm{r2}(B)}\mathrm{Hg}_{(B)\mathrm{int}}\qquad\text{Eq. 25}$$

$\mathrm{Hg}_{(b)\,\mathrm{ext}}$ = Hg on ecosystem group B      [ng/m³]

$\mathrm{Hg}_{(b)\,\mathrm{int}}$ = Hg inside ecosystem group B      [ng/m³]

$r_{\mathrm{r1}}$ = release rate of external Hg      [m³/s]

$r_{\mathrm{r2}}$ = release rate of internal Hg      [m³/s]

$$R_{\mathrm{M}(B)}=\sum_{B=0}^{m}\square r_{\mathrm{m}(B)}(\mathrm{Hg}_{(B)\mathrm{ext}}+\mathrm{Hg}_{(B)\mathrm{int}})\qquad\text{Eq. 26}$$

$r_{\mathrm{m}}$ = mortality rate of ecosystem group B      [m³/s]

| Nr | ecosystem group | Hg species | parameter | Description | value | unit | Source |
|----|-----------------|------------|-----------|-------------|-------|------|--------|
| 1 | diatoms | Hg²⁺ | $r_u$ | uptake rate | 3.95E-08 | s⁻¹ | *A* |
| 2 | | | $r_r$ | release rate | 6.58E-04 | s⁻¹ | *A+B* |
| 3 | | MeHg⁺ | $r_u$ | uptake rate | 4.84E-08 | s⁻¹ | *A* |
| 4 | | | $r_r$ | release rate | 8.40E-06 | s⁻¹ | *A+B* |
| 5 | flagellates | Hg²⁺ | $r_u$ | uptake rate | 1.87E-08 | s⁻¹ | *A* |
| 6 | | | $r_r$ | release rate | 3.11E-04 | s⁻¹ | *A+B* |
| 7 | | MeHg⁺ | $r_u$ | uptake rate | 1.82E-08 | s⁻¹ | *A* |
| 8 | | | $r_r$ | release rate | 8.40E-06 | s⁻¹ | *A+B* |
| 9 | cyanobacteria | Hg²⁺ | $r_u$ | uptake rate | 4.46E-08 | s⁻¹ | *A* |
| 10 | | | $r_r$ | release rate | 7.43E-04 | s⁻¹ | *A+B* |
| 11 | | MeHg⁺ | $r_u$ | uptake rate | 4.34E-08 | s⁻¹ | *A* |
| 12 | | | $r_r$ | release rate | 8.40E-06 | s⁻¹ | *A+B* |
| 13 | zooplankton | Hg²⁺ | $r_u$ | uptake rate | 1.94E-10 | s⁻¹ | *Tsui and Wang, 2004* |
| 14 | | | $r_{re}$ | external release rate | 6.94E-06 | s⁻¹ | *Tsui and Wang, 2004* |





| 15 | | | $r_{ri}$ | internal release rate | 5.79E-07 | s$^{-1}$ | *Tsui and Wang, 2004* |
|---|---|---|---|---|---|---|---|
| 16 | | MeHg$^+$ | $r_u$ | uptake rate | 2.56E-10 | s$^{-1}$ | *Tsui and Wang, 2004* |
| 17 | | | $r_{re}$ | external release rate | 2.32E-07 | s$^{-1}$ | *Tsui and Wang, 2004* |
| 18 | | | $r_{ri}$ | internal release rate | 5.80E-08 | s$^{-1}$ | *Tsui and Wang, 2004* |
| 19 | fish | Hg$^{2+}$ | $r_u$ | uptake rate | 3.88E-12 | s$^{-1}$ | *Pickhardt et al, 2007* |
| 20 | | | $r_{re}$ | external release rate | 3.47E-07 | s$^{-1}$ | *Pickhardt et al, 2007* |
| 21 | | | $r_{ri}$ | internal release rate | 6.45E-07 | s$^{-1}$ | *Pickhardt et al, 2007* |
| 22 | | MeHg$^+$ | $r_u$ | uptake rate | 1.00E-11 | s$^{-1}$ | *Pickhardt et al, 2007* |
| 23 | | | $r_{re}$ | external release rate | 2.30E-07 | s$^{-1}$ | *Pickhardt et al, 2007* |
| 24 | | | $r_{ri}$ | internal release rate | 2.30E-09 | s$^{-1}$ | *Pickhardt et al, 2007* |
| 25 | default | Hg$^{2+}$ | $\varepsilon$ | transfer efficiency | 0.45 | [] | *Pickhardt et al, 2007* |
| 26 | | MeHg$^+$ | $\varepsilon$ | transfer efficiency | 0.97 | [] | *Tsui and Wang, 2004* |
| 27 | fish | Hg$^{2+}$ | $\varepsilon$ | transfer efficiency | 0.158 | [] | *Wang and Wong, 2003* |

*Table 5: Overview of bioaccumulation parameters. External variables taken from the ecosystem model ECOSMO such as mortality ($r_m$) and feeding rates ($r_f$) are given in Table 1. The abbreviated phytoplankton references are, A: "Mason et al, 1996; Menden-Deur and Lessard, 2000; Olenina et al., 2003"; and B: "Pickhardt and Fisher, 2007; Nfon et al., 2009"*

### 2.3.5 Benthic-pelagic coupling

Following the sediment modelling concept by Daewel (2019), we implemented a simple two-layer sediment system, where the first layer interacts with the lowest water column grid cell and the second layer represents a permanent sink.

### Sedimentation

Sedimentation occurs due to the settling of Hg bound to particles and detritus. The sedimentation flux $F_s$ is calculated using a sedimentation velocity $w_d$ of 5 m/day for Hg bound to particles (POC) (Daewel and Schrum, 2013) (Eq. 27).

$$F_s = w_d Hg^{2+}_{POC} \qquad \text{Eq. 27}$$

F$_S$ = sedimentation flux  [ng/s m$^{-2}$]
Hg$^{2+}_{POC}$ = particulate mercury concentration in water  [ng/m³]
w$_d$ = sinking velocity  [m/s]

### Resuspension

Re-suspension $F_r$ is triggered by a critical ocean current velocity of $U$ 0.01 m/s. In case a critical current velocity is reached no sedimentation takes place and a resuspension rate $r_{res}$ of 25 [day$^{-1}$] is used to release Hg$^{2+}_{(s)}$ from the first sediment layer into the lowest water grid cell (Eq. 28). Depending on the depth (< 1m) of the lowest grid cell and current velocity, resuspension can also directly affect the second lowest water grid cell.





$$F_r = r_{res}\, Hg^{2+}_S \qquad \text{Eq. 28}$$

$F_r$ = resuspension flux          [ng/s m$^{-2}$]
$r_{res}$ = resuspension rate         [s$^{-1}$]
$Hg^{2+}_s$ = mercury concentration in sediment    [ng/m²]

**_Burial_**
$Hg^{2+}$ and MMHg$^+$ in the first layer are constantly transported to the second layer which represents a permanent sink in the model. The burial flux $F_b$ is based on a constant burial rate of $k_{bur}$ = 1.0E-04 [day$^{-1}$] (Eq. 29) (Table 4).

$$F_b = r_{bur}\, Hg^{2+}_S \qquad \text{Eq. 29}$$

$F_b$ = burial flux           [ng/s m$^{-2}$]
$r_{bur}$ = burial rate          [s$^{-1}$]
$Hg^{2+}_s$ = mercury concentration in sediment    [ng/m²]

**_2.3.6 Air-sea exchange_**

Air-sea exchange of elemental Hg$^0$ is one of the most important processes in the global Hg cycle. Here, we use the approach of Kuss et al. (2009; 2014) which is based on the Henry's Law constant $H$ by Andersson et al. (2008) to determine the equilibrium between Hg$^0$ in water $Hg^0_{(aq)}$ and air $Hg^0_{(air)}$ (Eq. 30). Next, the transfer velocity for CO$_2$ $k_{600}$ is approximated using a quadratic parametrization depending on 10m wind speed $U_{10}$ (Eq. 31). We then calculate the transfer

velocity $k_w$ for Hg$^0$ by scaling $k_{600}$ using the temperature $T$ and salinity $S$ dependent diffusivity of Hg$^0$ in water (Eqs. 32 to 35) (Kuss et al., 2014). The actual inter-compartmental Hg$^0$ flux $F_{Hg}$ is then calculated based on surface concentrations in the adjacent compartments (Eq. 36). The air-sea exchange is also applied for DMHg. However, the CMAQ-Hg model does not consider DMHg yet. Hence, the atmosphere is only a sink for DMHg which is instantaneously transformed into Hg$^{2+}$ (Niki et al., 1983) and it's fate is currently not explicitly resolved.


| | | |
|---|---|---|
| Eq. 30 | $H_{Hg} = e^{\left(\frac{-2404.3}{T} + 6.915\right)}$ | Anderssen et al., 2008 |
| Eq. 31 | $k_{600} = 0.222\, U^2_{10}\, 0.333\, U_{10}$ | Nightinggale et al., 2000 |
| Eq. 32 | $Sc_{25} = -0.0398\, T^3 + 3.3910\, T^2 - 118.02\text{T} + 1948.2$ | Kuss et al., 2014 |
| Eq. 33 | $Sc_0 = -0.0304\, T^3 + 2.7457\, T^2 - 118.13\text{T} + 2226.2$ | Kuss et al., 2014 |





Eq. 34     $Sc = \dfrac{Sc_{35}S + Sc_0(35-S)}{35}$          Kuss et al., 2014

Eq. 35     $k_w = k_{600}\sqrt{\dfrac{Sc}{600}}/360000$          Kuss et al., 2014

Eq. 36     $F_{Hg} = \dfrac{Hg^0_{(air)} - H_{Hg}*Hg^0_{(aq)}}{k_w}$          Schwarzenbach et al, 2003

|  |  |  |  |
|---|---|---|---|
| $H_{Hg}$ | = Henry's Law constant | [] |
| T | = water temperature | [°C] |
| S | = salinity | [psu] |
| $Sc_{35}$ | = Schmidt number for salt water | [] |
| $Sc_0$ | = Schmidt number for fresh water | [] |
| $k_{600}$ | = transfver velocity of $CO_2$ | [cm/h] |
| $k_w$ | = transfer velocity of Hg | [cm/h] |
| $Hg^0_{(air)}$ | = $Hg^0$ concentration in air | [ng/m³] |
| $Hg^0_{(aq)}$ | = $Hg^0$ concentration in water | [ng/m³] |
| $F_{Hg}$ | = net $Hg^0$ flux from atmosphere to water | [ng m$^{-2}$ h$^{-1}$] |

*560*
*565*

### 2.3.7 Technical implementation

As basis for the presented model development we build upon the setup used for the earlier published inorganic marine Hg model MERCY (Bieser and Schrum, 2016). All processes are implemented as stand-alone routines which are called from a main driver function containing several time loops (Figure 5). Data for the wet cells (pela-
gic) are stored in vector form to overhead and data for sediments (benthic) and the lowest atmospheric layer are stored in 2d fields. Input data (Table 1) is read directly during run time from binary ECOSMO output as hourly mean values. This approach was chosen because there is no feedback from the Hg chemistry on the physical and biological models, and it allows us to reduce the computational costs of running the
marine Hg model. All output files are created with daily mean values and saved in netCDF format using the IO-API interface (Byun and Schere, 2016; IO-API). The model is set up in a way that it runs for a single year using the last output timestep of the previous year as initial condition. For this initial model evaluation, we run MERCY for 17 years from 2000 to 2016.

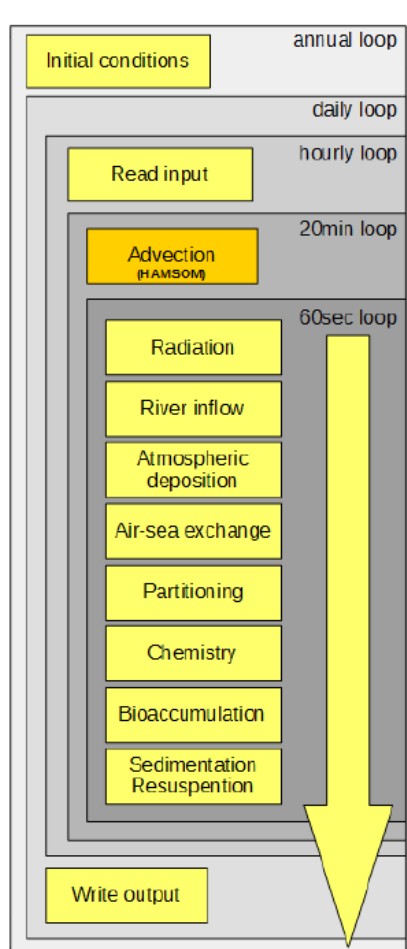


*Figure 5: Schematic overview of the MERCY model routines and main time loop.*





## 3 Model evaluation

We determine the model performance  in reproducing observed concentrations and dynamics (e.g. variability and seasonality)
of individual Hg species. Based on this analysis we identify the processes and parameters responsible for the model error.
The model is not specifically calibrated to the area of application, the North and Baltic Seas. It is built on the current under-
standing of mercury cycling in the ocean and should be generally applicable. Major factors that need to be considered before
applying the MERCY model to other regions are: (1) partitioning coefficients to organic material (OM) as the type of OM
varies regionally, (2) the parametrization for biogenic reduction as the values presented here are based on cyanobacteria in
the Baltic Sea, (3) the uptake and release rates for bioaccumulation which might not be representative for other regions and
(4) the used ecosystem model needed to drive MERCY.

### 3.1 Statistics

Because there are no established quality criteria for marine models we use   criteria commonly used for  for evaluation of at-
mospheric CTMs (Derwell et al., 2010; Thunis et al., 2012; 2013; Carnevale et al., 2014). We start by comparing the ob-
served and predicted means (Eq. 37) using daily model averages in the corresponding 10x10km² grid cell to the observation.
As statistical metrics, we use bias (Eq. 38), error (Eq. 39), standard deviation (Eq. 40, 41), and correlation coefficient (Eq.
42) to evaluate systematic error, random error, amplitude error, and phase error. However, for most Hg species the observa-
tions lack the temporal coverage to determine the phase error. Moreover, we use the centered root mean squared error
(CRMSE) because it allows us to distinguish between systematic error (bias) and random error (CRMSE) (Eq. 43) (Carne-
vale et al., 2014). For our analysis, we normalize the statistical metrics to get concentration-independent values and allow for
better comparability between different Hg species.

$$\text{mean:} \qquad O = \frac{1}{N} \sum_{i=0}^{N} O_i \quad P = \frac{1}{N} \sum_{i=0}^{N} P_i \qquad \text{(Eq. 37)}$$

$P_i$  = Predicted value from the model

$O_i$  = Observed values from measurement

$N$  = Sample size / number of observations

$i$  = index





normalized mean bias:

$$NMB = \frac{P - O}{O} \qquad \text{(Eq. 38)}$$

normalized centered root mean squared error:

$$NCRMSE = \frac{\sqrt{\frac{1}{N}\sum_{i=0}^{N}\{(O_i - O) - (P_i - P)\}^2}}{O} \qquad \text{(Eq. 39)}$$

standard deviation:

$$\sigma_O = \sqrt{\frac{1}{N}\sum_{i=1}^{N}(O_i - O)^2} \quad \sigma_P = \sqrt{\frac{1}{N}\sum_{i=1}^{N}(P_i - P)^2} \qquad \text{(Eq. 40)}$$

normalized mean standard deviation:

$$NMSD = \frac{\sigma_P - \sigma_O}{\sigma_O} \qquad \text{(Eq. 41)}$$

correlation correlation (r):

$$r = \frac{\frac{1}{N}\sum_{i=1}^{N}(O_i - O)(P_i - P)}{\sigma_O \sigma_P} \qquad \text{(Eq. 42)}$$

root mean square error:

$$RMSE^2 = CRMSE^2 + bias^2 \qquad \text{(Eq. 43)}$$

Finally, we use additional quality criteria to determine model performance. Firstly, the percentage of model values within a factor of 2 (FAC2), which gives an easy-to-understand estimate of the model quality (Eq. 44). We argue that model values within a factor of 2 are within the combined uncertainty. The uncertainty consists of the measurement uncertainty, the sampling uncertainty when comparing observations with time (24h) and space (100km²) integrated model grid cells

(Schutgens et al., 2010) and the error propagation in the biogeochemical modelling framework. We estimate the sampling error to be in the range of 30% and the measurement error to range from 20% ($Hg^0$ and $Hg_T$) to 50% (MeHg).

$$\text{Factor of 2} \quad FAC2 = \frac{1}{N}\sum_{i=1}^{N} n_i \qquad \text{(Eq. 44)}$$

$$\text{with} \quad k_1 \quad \text{for} \quad 0.5 < \frac{P_i}{O_i} < 2 \quad \text{and else } n_i = 0$$

Secondly, we use the more technical model quality objective (MQO) as defined by Carnevale et al., (2014). The MQO (Eq.

45) relates the root mean squared error (Eq. 46) to the root mean squared uncertainty (Eq. 47). The MQO can be interpreted as follows: For $MQO < 0.5$ on average the model values lie within the measurement uncertainty and thus the model cannot be improved upon unless more precise observations become available. For $MQO > 1.0$ the model error is on average larger than the measurement uncertainty but the model may be closer to the 'true' environmental value than the observations. Thus, the aim is to achieve an $MQO < 1.0$. Moreover, we determine model performance criteria for NMB,

NMSD, and RMSE as proposed by Carnevale et al. (2010) (Eqs. 49-51).





$$\text{model quality objective:} \qquad \text{MQO} = \frac{1}{2}\frac{\text{RMSE}}{\text{RMS}_U} \qquad \text{(Eq. 45)}$$

$$\text{root mean squared error:} \qquad \text{RMSE} = \sqrt{\frac{1}{N}\sum_{i=1}^{N}\left(P_i - O_i\right)^2} \qquad \text{(Eq. 46)}$$

$$\text{root mean squared uncertainty:} \qquad \text{RMS}_U = \sqrt{\frac{1}{N}\sum_{i=0}^{N}U^2} \qquad \text{(Eq. 47)}$$

$U$ = measurement uncertainty


$$\text{model performance criterion:} \qquad MPC_{NMB} \vee \frac{2U}{O} \qquad \text{(Eq. 48)}$$

$$\text{model performance criterion:} \qquad MPC_{NMSD} \vee \frac{2U}{\sigma_O} \qquad \text{(Eq. 49)}$$

$$\text{model performance criterion:} \qquad MPC_{RMSE} \leq 1.0 \qquad \text{(Eq. 50)}$$

These quality criteria have been developed for atmospheric pollutants like ozone, nitrogen oxides, and fine particles which have been studied and modelled for decades. For modelling of Hg in the marine environment, the observations are still very limited compared to that of pollutants in the atmosphere. This reflects on the ability to use these criteria and we therefore do not expect the MERCY model to meet the criteria at this point. However, we define these as our future goal for marine Hg modelling.

## 650 3.2 Model domain (North Sea and Baltic Sea)

Here, we evaluate the model for the North and Baltic Sea in northern Europe (model domain shown in Figure 6). This area was chosen for model evaluation as it covers a large range of different physical and biological conditions: The Baltic Sea (Figure 6; marine regions 8-15) is an enclosed shelf sea with a surface area of 377 000 km². It is connected to the North Sea (marine regions 1-5) via the shallow Kattegat and Skagerrak (marine regions 6-7) in the southwest. It is a brackish water

body strongly influenced by freshwater inflow and it covers a salinity range from <2 PSU in the north that increases towards the southwest reaching up to 35 PSU in the transition zone between the North- and Baltic Sea. The central Baltic has several deep basins reaching a depth of 460 in the Landsort deep in the Central Baltic Sea (Figure 6b). It exhibits strong stable stratification with more saline, and in parts anoxic, deep water, resulting from an estuarine circulation system with upper layer outflow of fresh water and lower layer saline inflow. Every few years, large quantities of oxic and saline waters are transpor-

ted from the North Sea to the Baltic Sea during so-called Major Baltic Inflows (MBI). During the simulation period 2000 to 2015 three MBIs occurred, one of these was an especially strong event during the winter of 2014-2015 (Fischer et al., 1996;





Lehmann et al., 2015). In the northern part of the Baltic, the Bothnian Sea and the Bothnian Bay are seasonally covered by ice, possibly leading to accumulation of Hg from rivers during winter due to the suppression of $Hg^0$ evasion. Finally, the Baltic Sea is a system with cyanobacteria, which make it an interesting study area as they have been shown to actively reduce

$Hg^{2+}_{(aq)}$ (Kuss et al., 2015). Moreover, cyanobacteria can lead to pronounced early spring / late summer biomass blooms affecting bioaccumulation (Soerensen et al., 2016).

The North Sea has a surface area of 575 000 km² and is connected to the Atlantic Ocean at its northern border and via the English Channel to the South. It is a shallow shelf ocean that is well mixed during autumn and winter and it experiences frequent resuspension events during autumn and winter storms. The southern North Sea is characterized by strong tidal mixing

and thus water masses are well mixed and sediments are resuspended regularly within the tidal cycle. It is an area of high primary productivity and an important fishing ground. Thus, it is an important study area for Hg methylation and bioaccumulation.

Due to the close vicinity to the coast and national monitoring programs, there is a comparably large number of Hg observations available for both the North Sea and the Baltic Sea. However, the data on Hg are still sparse in some areas, especially

regarding Hg speciation, which is a major obstacle for model evaluation.

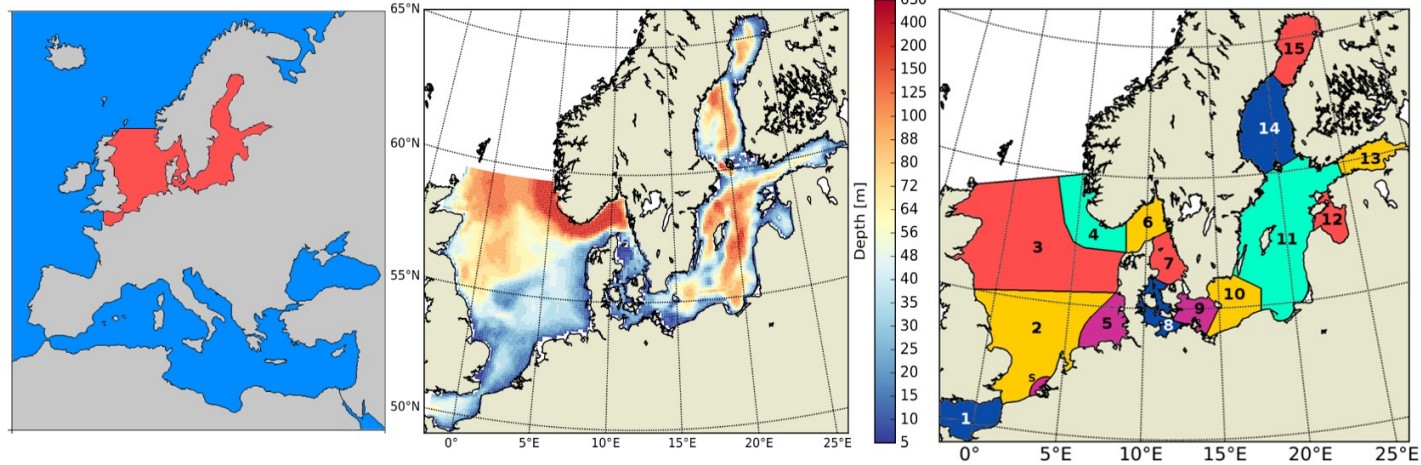

*Figure 6: Left: COSMO-CMAQ-Hg atmospheric model domain with North- and Baltic Sea highlighted. Middle: MERCY marine model domain and topography. Right: Marine regions: (1) English Channel, (S) Schelde Estuary, (2) Southern North Sea, (3) Northern North Sea, (4) Norwegian Trench, (5) German Bight, (6) Kattegat, (7) Skaggerak, (8) Belt Sea, (6-8) Swedish west coast, (9) Arkona Sea, (10) Bornholm Sea, (11) Gotland Sea or Baltic Proper, (12) Bay of Riga, (13) Neva Bay,*

*(14) Bothnian Sea, (15) Bothnian Bay.*



### 3.2.1 Forcing data

To generate the necessary forcing data (Table 1) to run the MERCY model, we used the four models described in Section 2.1.
For the atmosphere, COSMO-CLM was run on a regional domain for Europe driven by ERA interim re-analysis data (Berris­ford et al., 2011). The atmospheric model domain covers the entire European land mass, including North Africa and western Russia with a resolution of 24x24km and 35 vertical layers (Fig. 6a). The calculated meteorology is then used as forcing for the atmospheric CTM CMAQ-Hg which is set up for the same domain and resolution (Buyn and Schere, 2006; Zhu et al., 2015; Bieser et al., 2016). CMAQ-Hg uses boundary concentrations for Hg by an ensemble of the global Hg models GEOS-
Chem, GLEMOS, ECHMERIT, GEM-MACH-Hg (Travnikov et al., 2018) and all other relevant trace gases from the global CTM MOZART (Horowitz et al., 2003). Emissions for the year 2010 were created with the SMOKE-EU emission model (Bieser et al., 2011). Hg emissions are based on the AMAP emission inventory (AMAP, 2010). This is a similar setup as used in previous studies (Bieser and Schrum 2016). For computational reasons, we calculated only one year (2010) of atmospheric Hg concentration and deposition fields. These were used as boundary conditions for the marine Hg model for all years of the
simulation. The ocean-ecosystem model HAMSOM-ECOSMO was run on a model domain covering the entire Baltic Sea and the North Sea with open boundaries in the English Channel and at 63° North, where the North Sea is connected to the At­lantic (Fig. 6b). The resolution of the model is about 10x10 km$^2$ (spherical grid) with 20 layers and a maximum water depth of 630m. The vertical resolution is 5m for the four uppermost layers with a bottom layer depth of 250m.

### 3.2.2 Initial conditions

As initial conditions, we interpolated observations in water, biota, and sediment using a traditional Krieging methodology to produce realistic initial starting conditions (mostly the pronounced vertical gradient) and minimize the spin-up time required (Cressie, 1990). The observational Hg data were retrieved from the database of the German Federal Maritime and Hydro­graphic Agency (MARENET, 2020). We run the model using initial conditions multiplied by factors of 0.5 and 2.0 and tested the time necessary for the two runs to converge. For our model domain, which is a relatively small and in parts enclosed shelf
sea area, the model runs started to converge already after a few years in the water column but took several years for Hg in sediments and biota (expecially in higher trophic levels). For this study, we used a spin-up time of 30 years to reach realistic initial conditions for the production runs.

### 3.2.3 Boundary conditions

The chosen domain, including only the North Sea and Baltic Sea, has only a very small open boundary. The English Channel
in the southwest, which forms a narrow connection to the Atlantic Ocean and the wider opening in the Northern channel. The North Sea in the north of the domain, which receives the most of the Atlantic inflow is connected to the open Atlantic Ocean at the shelf break. This region is characterized by an outflow in the eastern part and inflow in the western part. At the open boundaries, we prescribe constant Hg concentrations using 1.0 pM Hg$_T$ for the North Atlantic and 3.0 pM Hg$_T$ in the English Channel (Cossa et al., 2018; Leemakers et al., 2001).




### 3.2.4 River loads


River loads are taken from OSPAR and HELCOM reports and the Norwegian Tilførsel program (Green et a., 2011, HEL-COM, 2007; 2011). We implemented rivers using monthly load data in the North Sea and annual values for the Baltic Sea as described in Bieser and Schrum (2016). The annual inflow of Hg through rivers is 1100 kg/a for the Baltic and 2800 kg/a for the North Sea. In the North Sea, the largest fluxes are from the rivers Elbe (1160 kg/a) and Rhine (800 kg/a).

### 3.2.5 Deposition of $Hg^{2+}$ and atmospheric $Hg^0$


Dry and wet Hg deposition is read in as hourly totals from CMAQ netCDF output files. The deposited $Hg^{2+}_{(g)}$ and $Hg^{2+}_{(p)}$ is added to the dissolved $Hg^{2+}_{(aq)}$ species assuming instant dissolution of atmospheric particles. In CMAQ, the exchange of $Hg^0$ is set to zero for all grid cells with land use category water to avoid a doubling of the air-sea exchange calculation in the atmospheric model.



## 3.3 Observational data

For the model performance we start by evaluating total Hg ($Hg_T$) concentrations in the water column. We then look at the individual species, elemental $Hg^0$ and organic MeHg. Next, we evaluate the model skill in reproducing Hg concentrations in biota. For this, we compare Hg and MeHg loads in phytoplankton and zooplankton, and finally total Hg in fish ($Hg_{Fish}$).

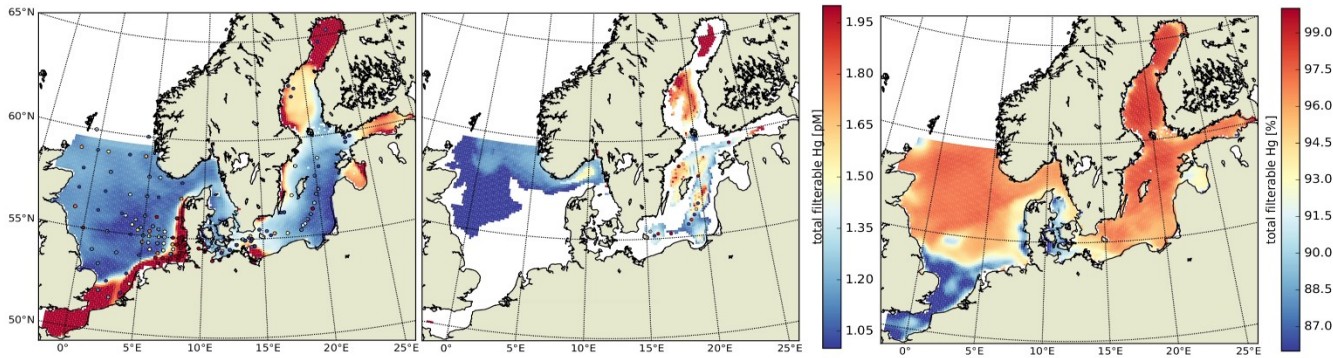

*Figure 7: Annual averages: a) $Hg_T$ concentrations in the top 50m with superimposed observations (Kuss et al., 2017; Soerensen et al. 2018; MARNET, 2020), b) $Hg_T$ concentrations below 50m with superimposed observations, c) average fraction of filterable $Hg_{Filt}/Hg_T$.*

### 3.3.1 Total Hg in water

The available $Hg_T$ observations cover offshore and coastal areas in the North and Baltic Sea. $Hg_T$ has been measured as both unfiltered (Soerensen et al. 2018) and filterable fraction $Hg_{Filt}$ (Kuss et al., 2017; MARNET, 2020). In MERCY, $Hg_{Filt}$ corresponds to the sum of 9 species, namely $Hg^0_{(g)}$, $Hg^{2+}_{(aq)}$, $Hg\text{-}DOM_{(aq)}$, $HgS\text{-}DOM_{(aq)}$, $MMHg^+_{(aq)}$, $MMHg\text{-}DOM_{(aq)}$, $DMHg_{(g)}$, and $HgS_{(s)}$ (Table 2). $Hg_T$ additionally includes $Hg_P$, which are comprised by the two particulate species $Hg\text{-}POC_{(s)}$ and $MeHg\text{-}POC_{(s)}$. In our model $Hg_{Filt}$ makes up about 95% of $Hg_T$ on average (Figure 7c). $Hg_P$ only plays a significant role (>5% on annual average) in the southern North Sea and the Wadden Sea. Especially in the Wadden Sea observed $Hg_T$ concentrations are extremely high with values ranging from 6 to 117 pM. For the model performance evaluation, we removed measurements taken in the Wadden Sea and other areas not resolved in our model setup (e.g. the area between coast line and barrier islands and lagoons in the Baltic Sea). As depicted in Figure 7a, virtually no observations are from regions where particles play a major role. Thus, for simplification we will use the term $Hg_T$ to refer to all of these observations, but compare them to concentrations of the respective model species.

In the North Sea we use 435 measurements of $Hg_T$ sampled between 2007 and 2011 (MARNET, 2020). The samples are taken over the entire year but BSH (The German Federal Maritime and Hydrographic Agency) sampling focuses mostly on the German exclusive economic zone, yet also includes a few years with data for the greater North Sea. Finally, all measurements are surface samples (0-20m) which is due to the shallow nature of the North Sea. For the Baltic Sea, there are



111 observations from the MARNET database (MARNET, 2020), 168 observations from three cruises in March 2014, March
2015, and July/August 2015 which cover the southern part of the Baltic Sea from the Belt Sea to the Gotland Deep (Kuss et
al., 2017), and 90 observations from three cruises in July and August of 2015 and 2016 which in addition includes
observations on the Bothnian Sea and Bothnian Bay (Soerensen et al., 2018). Figure 7a,b depict all $Hg_T$ observations used for
model evaluation.

755

### 3.3.2 Individual marine species: $Hg^0$ and $MMHg^+$

For the evaluation of $Hg^0$, we use 580 measurements from four Baltic Sea cruises in February, April, July, and November
2006 (Kuss et al., 2014). This dataset allows us to investigate the seasonality of red-ox reactions. For $MMHg^+$ we were able
to obtain 310 measurements from six cruises in 2014, 2015, and 2016 covering coastal and offshore areas of the Baltic Sea
(Kuss et al., 2017; Soerensen et al., 2017; 2018). For 160 of these both MeHg and $Hg_T$ were available which enable a relative
evaluation of methylated fraction $M_{frac}$ = MeHg/HgT. For the North Sea, no $Hg^0$ or $MMHg^+$ observations are available at all
with the exception of 9 MeHg measurements from 1999 in the English Channel and the Schelde Estuary which we used to set
the $MMHg^+$ boundary conditions in the English Channel (Leemakers et al., 2001). Thus, we are forced to limit the model
evaluation for $Hg^0$ and $MMHg^+$ to the Baltic Sea.


### 3.3.3 Hg in biota

Bioaccumulation in the marine biota is evaluated by comparing their total Hg and MeHg content to measured concentrations
in biota in the Baltic Sea (Nfon et al, 2009). For evaluation of fish total Hg we use $Hg_T$ concentration in muscle of 1166 her-
ring from coastal and offshore location in the Baltic Sea (Soerensen and Faxneld, 2020). As the biota measurements are in
wet weight and our model is in mg carbon dry weight, the ratio of mg carbon per mg biota of: 0.2 for diatoms, 0.33 for fla-
gellates and cyanobacteria, and 0.5 for zooplankton and fish was assumed (Sicko-Goad et al., 1984; Walve and Larsson
1999). This was combined with a conversion factor of dry weight to wet weight of 0.2 for phytoplankton, 0.16 for zooplank-
ton, and 0.1 for fish (Cushing 1958; Ricciardi and Bourget 1998). For phyto- and zooplankton, the model is compared to the
observed average, minimum and maximum concentrations, but due to limited data no seasonal or regional comparison was
possible. For fish we analyse Hg accumulation for five Baltic Sea regions ranging from the western Baltic to the Bothnian
Bay.





## 3.4 Model performance

### 3.4.1 North Sea (Hg$_T$ )

Figure 8 compares the frequency distribution of 435 Hg$_T$ measurements to the associated model values. It can be seen that the

majority of observations lie between 1 and 3 pM, which is captured well by the model. However, the observed high values between 5 and 10 pM cannot be reproduced. We argue that these are due to input from the coastal area (e.g. major rivers, Wadden Sea) not included as input to the model in the current river discharge scenario.

HgT concentrations in the North Sea do not exhibit a pronounced seasonality and the observed variability is driven by a strong land-sea gradient along the European coastline where Hg from rivers is transported north-eastwards from the English

Channel by the Coriolis force (Figure 7a). For the analysis, we split the North Sea Hg observations into two groups: (1) The Elbe Estuary (N=366) and (2) the open North Sea including a few observations near the remaining coastline (N=69). Due to the significant Hg inflow from the Elbe (1164 t/a), Hg concentrations are higher in the Elbe estuary with a mean concentration of 3.44 pM (Table 6). The model is able to reproduce the observed average (NMB = -21%), but has a better agreement with the median values (-12%). In this region, random and amplitude errors are dominant. This is indicative of subgrid dy-

namics and our inability to resolve the seasonality of Hg from rivers stemming from the use of static river loads for the entire run (OSPAR, 2016). However, with 70% of model values within a factor of 2 of the observations and an MQO = 1.48 the model is still close to our quality goal.

In the less dynamic open North Sea, the model performs better (FAC2 = 84%, MQO = 1.22) (Table 6). The observed average of 1.92 pM is matched by the model (2.03 pM) and the bias is close to zero (NMB=6%). Nevertheless, due to the inhomo-

geneous distribution of observations, this value is not indicative of the actual background Hg concentrations in the open North Sea. We find, that Hg concentrations there are mostly in the range of 1.1 to 1.5 pM.

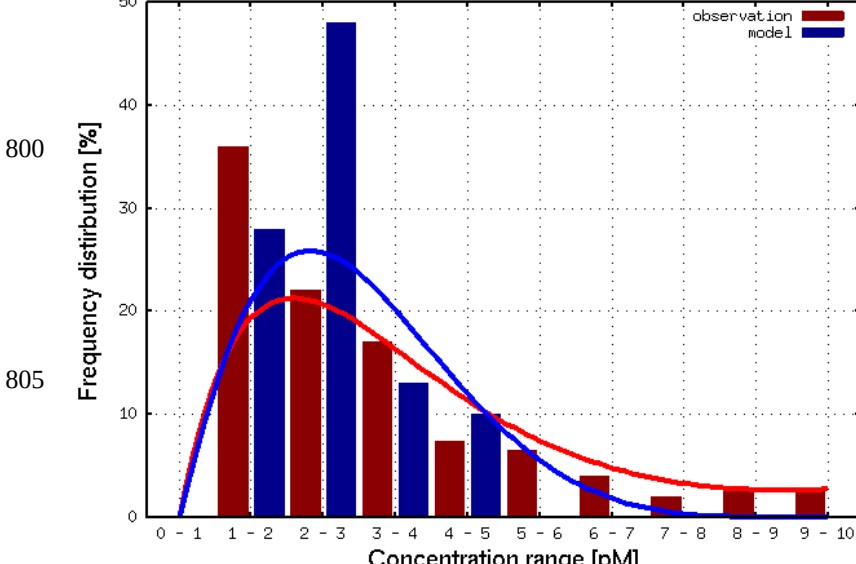

*Figure 8: Frequency distribution of observed (red) and modelled (blue) Hg$_{Filt.}$ Concnetrations for the North Sea (N=435).*




In summary, for $Hg_T$, the model is close to our quality criterion (MQO<=1.0). Improvements to the MQO could likely be achieved by increasing model resolution in the complex coastal regions and including more detailed input from rivers and particle resuspension at the European coastline. Especially for the Wadden Sea, a hydrodynamical model that can model the intertidal zone would be preferential.

| Elbe Estuary | obs [pM] | mod [pM] | NMB | NCRMSE | FAC2 | MQO | N |
|---|---|---|---|---|---|---|---|
| mean | 3.44 | 2.71 | - 0.21 | 0.66 | 70 % | 1.48 | 366 |
| median | 2.78 | 2.44 | - 0.12 | | | | |
| stdev | 2.16 | 0.82 | - 0.62 | | | | |
| **Open North Sea** | **obs [pM]** | **mod [pM]** | **NMB** | **NCRMSE** | **FAC2** | **MQO** | **N** |
| mean | 1.92 | 2.03 | 0.06 | 0.69 | 84 % | 1.22 | 69 |
| median | 1.68 | 1.74 | 0.03 | | | | |
| stdev | 0.80 | 0.67 | - 0.16 | | | | |

*Table 6: Regional model performance for $Hg_T$ in the North Sea. The evaluation is based on 435 measurements from the MARENET database (MARENET, 2020).*

### 3.4.2 Baltic Sea ($Hg_T$)

In the Baltics Sea, model performance for $Hg_T$ is similar to the North Sea (FAC2 = 70%, MQO = 1.28) with a low bias (NMB = -19%) and a high random error (NCRMSE = 102%) (Table 7). Unlike for the North Sea, the model predicts a pronounced seasonality with surface $Hg_T$ concentrations around 50% higher during March (Figure 9a) compared to August (Figure 9b) which is in line with observations from Kuss et al. (2017) taken in March and July/August (Figure 9). The two processes governing this are: (1) Stratification and particle settling in the central Baltic deep basins after the onset of primary production and (2) increased photoreduction and subsequent atmospheric exchange of $Hg^0$. Additionally, during winter higher atmospheric $Hg^0$ concentrations due to heating related emissions and a shallow planetary boundary layer reduce and sometimes even reverse the $Hg^0$ air-sea gradient. In the open Baltic Sea, Hg concentrations are mostly between 1.0 and 1.5 pM. In stratified areas, $Hg_T$ concentrations can drop down to 0.5 pM during summer. During autumn and winter mixing and upwelling can occasionally transport Hg from deeper waters upwards, sometimes leading to surface concentrations above 2 pM in some areas.

| Region | Depth | obs [pM] | mod [pM] | NMB | NCRMSE | FAC2 | MQO | N |
|---|---|---|---|---|---|---|---|---|
| **Baltic Sea (all)** | **0 – 250m** | **1.83** | **1.45** | **- 0.19** | **1.02** | **70%** | **1.28** | **336** |
| **Western Baltic** | 0 – 100m | 1.55 | 1.50 | - 0.03 | 0.97 | 72 % | 1.76 | 168 |
| **Central Baltic** | 0 – 100m | 1.39 | 1.39 | 0.00 | 0.81 | 73 % | 1.03 | 103 |
| **Central Baltic** | 100 – 250m | 2.38 | 1.51 | - 0.37 | 0.67 | 67 % | 1.02 | 50 |
| **Nothern Baltic** | 0 – 100m | 0.85 | 1.82 | 1.16 | 1.25 | 23 % | 2.01 | 15 |

*Table 7: Observed and modelled seasonal and regional of $Hg_{Filt.}$ concentrations in the Baltic Sea.*



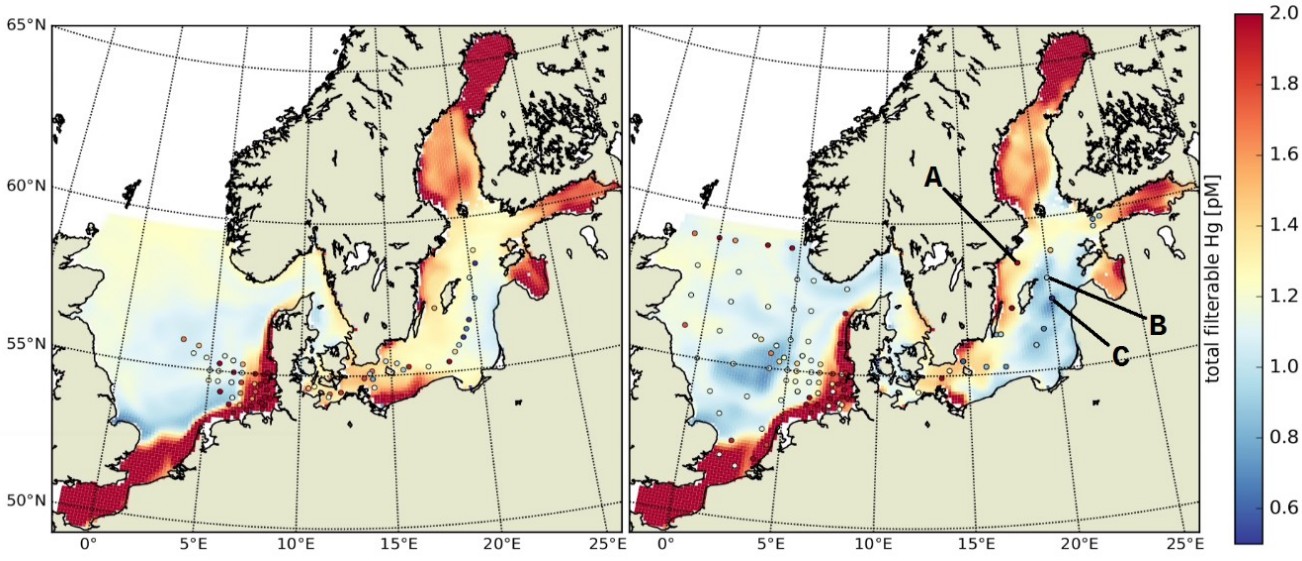


*Figure 9: Hg$_T$ surface concentrations for a) March and b) August. Dots indicate observations (Kuss et al., 2017; Soerensen et al., 2018; MARNET, 2020).*

For a more detailed analysis, we separated the Baltic Sea into three regions: (1) The western part, which includes the Belt, Arkona, and Bornholm Sea, (2) the Gotland Sea in the central Baltic, and (3) the northern part which includes the Bothnian

Sea and Bothnian Bay. Moreover, we evaluate the oxic surface/intermediate waters and the deep anoxic waters in the Gotland area separately (Table 7). It is seen, that the model is able to reproduce surface concentrations in the western and central areas with a bias close to zero. The model bias is larger in the deep basins but model performance is still comparable to the North Sea. Here, the low vertical resolution in the model setup below 100m will play a role. In the northern part, the model strongly overestimates Hg$_T$ concentrations. This overestimation was also seen in the Soerensen et al. (2018) model. Northern Baltic

rivers tends to be low in POC but rich in DOC compared to temperate rivers (McClelland et al. 2016; Soerensen et al. 2017) highlighted the importance of DOC flocculation at the point where river water encounters higher salinity water for the settling and removal of Hg in Bothnian Bay estuaries, something that is currently not included in our model.




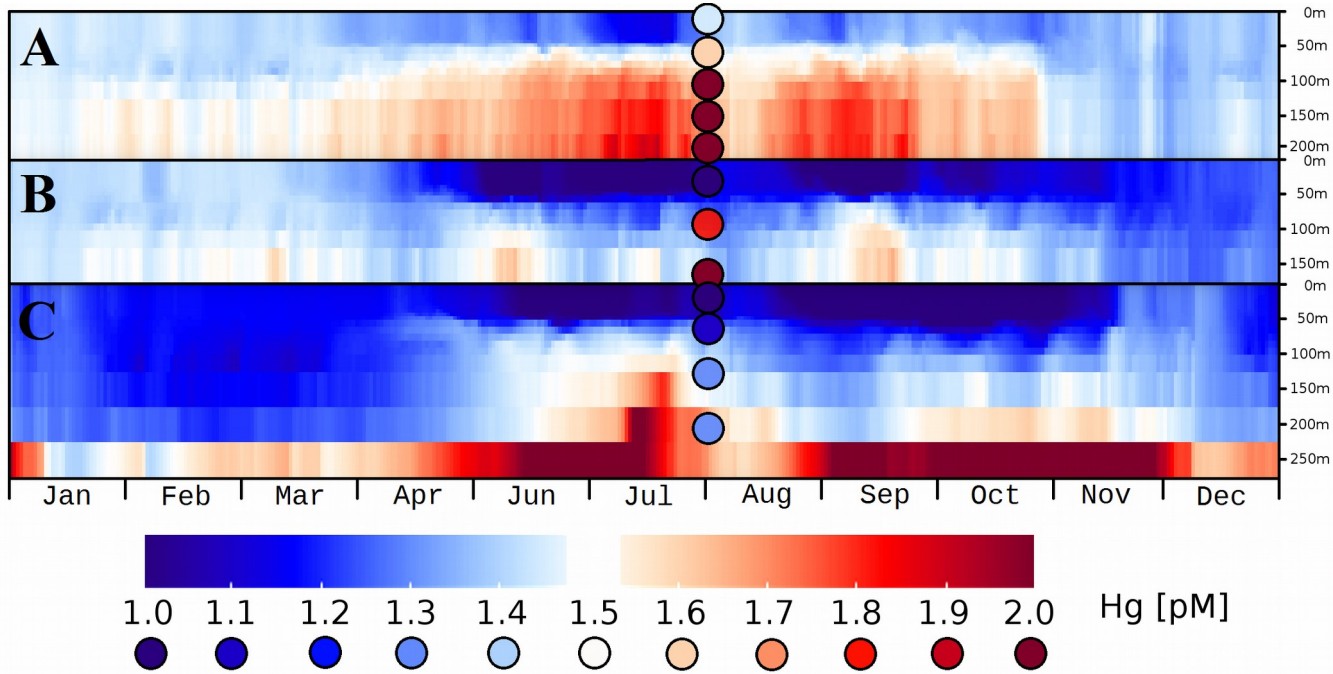

*Figure 10: Hg$_T$ seasonality of vertical profiles at three Baltic Sea locations (Fig. 9) for the year 2015 superimposd with ob-*
*servations. The x-axis gives 365 daily averages for the year 2015.*

Figure 10 depicts the seasonality and figure 11 three vertical profiles in the Gotland Sea. It is seen how quickly Hg concen-
trations can change in this region and, depending on physical drivers, how different the seasonality of vertical mixing can be.
At location A (Gotland Deep) Hg concentrations are around 1.5 pM for most of the year with a strong surface depletion (1
pM) during August and September. At location C, located at the opposite side of Gotland, the seasonality is reverse with the
highest concentrations (1.2-1.4 pM) during August and September and much lower concentrations (0.9 − 1.1 pM) throughout
the rest of the year.

In summary, our conclusion is similar to that of the North Sea, i.e. that better data on Hg inputs from rivers and a better resol-
ution of the physical processes in the domain seem the most promising options for improving model performance. Especially
in the Bothnian Bay, Hg cycling seems to be strongly influenced by terrestrial organic matter. In the central Baltic, we found
that typically used kd values around $\log(k_d) = 5.5$ are not sufficient to reproduce the observed Hg depletion in the surface wa-
ters. Here, as described in the methods section, we use a $k_d$ based on Tesan et al. (2020) which is an order of magnitude
higher and leads to improved correlation to observations (Table 4). In addition, a higher vertical resolution is advised as ver-
tical transport has proven to be an issue with the current model setup. Due to the low model resolution below 150 m, numer-
ical diffusion leads to an overestimation of mixing in the deep basins. Finally, for further model evaluation, it would be useful
to increase the seasonal coverage of observations in the area.



*Figure 11: Vertical Hg profiles in the central Baltic Sea observations (blue) (Soerensen et al., 2018) and model values (red)*
*for the three central Baltic deep basins given in Figure 9.*





### 3.4.3 Elemental mercury (Hg0)

In the marine environment, elemental Hg$^0$ makes up only a few percent of the total Hg$_T$. However, it is the species that determines the air-sea exchange and thus is the major driver for atmospheric long-range transport. With the oceans being the
largest Hg emitter into the atmosphere (roughly twice as large as current anthropogenic emissions), marine Hg$^0$ determines global transport patterns. Moreover, errors in modelled Hg$^0$ concentrations propagate to all other Hg species and lead to wrong estimates for the compartmental budgets. Thus, it is of utmost importance to correctly reproduce Hg$^0$ concentrations in surface waters. A detailed model study on Hg air-sea exchange in the North and Baltic Sea has been published using a previous model version (Kuss et al., 2014; Bieser and Schrum, 2015). The four main drivers of Hg$^0$ concentrations are:


1) The reducible fraction of Hg$^{2+}$ which is typically estimated to be 40% of the dissolved Hg$^{2+}_{(aq)}$.

2) The parametrization of biologically induced reduction processes.

3) The modelled photon flux and wavelength-dependent extinction in water impacting photolytic reduction.

4) Air-sea exchange parametrizations, especially during high wind speeds.


Due to the fast exchange between atmosphere and water, Hg$^0$ concentrations converge towards the equilibrium as described by Henry's Law constant (Andersson et al., 2008). Therefore, in shelf seas a change in the red-ox chemistry directly affects the total Hg$^T$ in the system. Due to the mixing in the coastal ocean, this impacts almost the complete water body. Moreover, the different reduction pathways produce a distinct seasonal pattern with Hg$^0$ concentrations ranging from as low as 5 pg/l
during winter up to peaks > 60 pg/l during cyanobacteria blooms. Thus because of the high intra annual variability the model needs to be evaluated against Hg$^0$ observations throughout the year, as good agreement for a single cruise does not imply good model performance throughout the year.

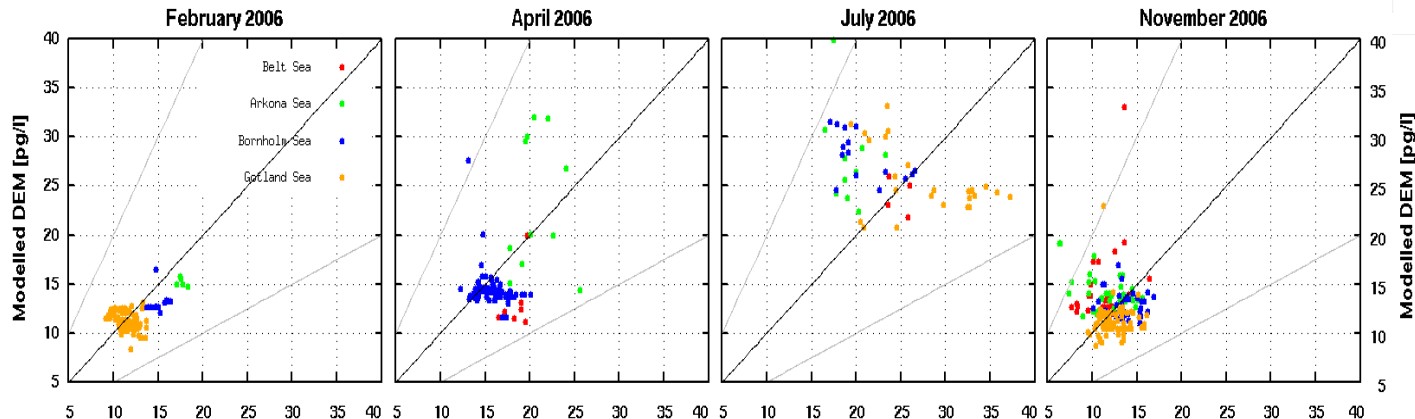

*Figure 12: Comparison of modeled and observed Hg$^0$ concentrations in surface waters for four cruises in the Baltic Sea (x-*
*axis: observations, y-axis: model values (Kuss et al., 2014).*



| obs mean | mod mean | NMB | NCRMSE | NMSD | FAC2 | N |
|---|---|---|---|---|---|---|
| 14.6 | 14.9 | 0.02 | 0.35 | 0.38 | 97% | 477 |
| **obs std** | **mod std** | **NMB$_{crit}$** | **RMSE** | **NMSD$_{crit}$** | **R** | **MQO** |
| 4.60 | 6.30 | | | | 0.60 | 0.84 |

*Table 8: Comparison of modelled and observed Hg$^0$ concentrations for four Cruises in the Baltic Sea in 2006 (Kuss et al., 2014).*

The observed annual average Hg$^0$ concentration for 580 measurements is 14.6 pg/l, the modelled value is 14.9 pg/l with a systematic error of 2% and a random error of 35%. The random error is largest in summer (NCRMSE=42%) and is due to the biogenic reduction which depends on cyanobacteria biomass. The model shows good correlation with observations (R=0.60) and is able to reproduce the observed variability (NMSD = 38%) (Table 8). All statistical metrics for Hg$^0$ are well inside model performance criteria and 97% of the model values are within a factor of 2 of the observations. The model quality objective is below 1.0 (MQO = 0.84). Thus, the model performance is, at least for the given model resolution, in the range
where further improvements hardly feasible (Carnevale et al., 2014; Schutgens et al., 2016).

We acknowledge, that the red-ox chemistry used is based on measurements in the Baltic Sea (Kuss et al., 2015). Thus, it needs to be investigated whether it shows equally good performance for other marine regions. We find, that the model performs similarly well throughout the year with the largest bias during summer where the dynamics driving biological and photolytic reduction lead to a higher variability of Hg$^0$ concentrations (Table 9).

| | obs mean | mod mean | NMB | NCRMSE | NMSD | N |
|---|---|---|---|---|---|---|
| **February** | 12.0 | 11.4 | - 0.05 | 0.28 | 0.41 | 130 |
| **April** | 16.5 | 15.2 | - 0.08 | 0.23 | 0.70 | 111 |
| **July** | 23.1 | 28.0 | 0.21 | 0.42 | 0.13 | 62 |
| **November** | 12.4 | 12.7 | 0.03 | 0.28 | 0.30 | 174 |

*Table 9: Seasonal breakdown of Hg$^0$ model performance.*

Figure 13 depicts the seasonality of a mean Hg$^0$ for the Baltic Sea. Moreover, the contribution of the four reduction reactions (1) chemical reduction, (2) photolytic reduction, (3) biogenic reduction, and (4) reductive methylation (Table 3) are given. We find, that the dark reduction is the dominant process, producing 55% of the total Hg$^0$ in the Baltic Sea and 70% in the North Sea. Photolytic reduction contributes 34% and biogenic reduction contributes 12% annually. However, from July to
middle August the photolytic reduction becomes dominant (>50%). When the cyanobacteria bloom starts, light penetration reduces significantly due to the increased marine particle load and until the end of November the biogenic reduction becomes the dominant process (Figure 13a). In contrast, as there are no cyanobacteria in the North Sea, photolytic reduction is dominant throughout the summer (Figure 13b). The reductive methylation reaction plays a negligible role for Hg$^0$ surface concentrations but can be a source for Hg$^0$ in deeper waters with a high MeHg fraction. It can be seen, that there is a background
Hg$^0$ concentration of about 5 to 15 pg/l in due to the chemical (dark) reduction process. During model development, we recognized a systematic error in the seasonality (overestimation during summer and underestimation during winter) that could





be resolved by introducing a temperature dependency of the chemical reduction reaction, a process which was detectable in the observations by Kuss et al. (2015) (Eq. 8, Section 2.3.1). For the photolytic reaction, we found that it is important to val-

idate the radiation fields. Testing the model using different radiation fields resulted in a change of the annual net $Hg^0$ production of > 10%. The main driver here is cloud coverage which is a particularly uncertain state variable in meteorological models. Moreover, we want to note that photolysis rates from observations and incubation experiments are solely reported based on the photolytically active radiation. Due to the highly wavelength selective light extinction it would be favourable to parameterize photolysis using the actual wavelengths absorbed by Hg. Finally, the biogenic reduction term in the model is driven only by the concentration of cyanobacteria. This creates the observed late summer / early autumn $Hg^0$ peak. Allowing other

phytoplankton species in the model to reduce $Hg^{2+}$ lead to unrealistically high concentrations during the spring flagellate bloom.

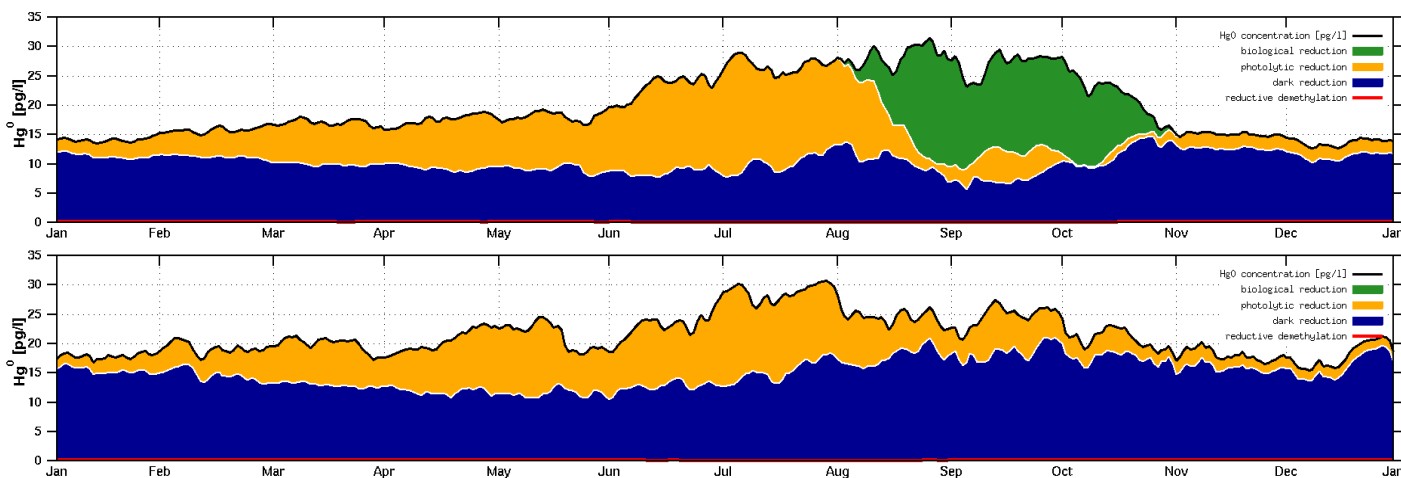

Figure 13: Annual profile of mean $Hg^0$ concentration in the Baltic Sea. The colored areas indicate the contribution of indi-

vidual reduction pathways (R1,R3,R20 Table 3).





### 3.4.4 Methylmercury (MMHg$^+$ and DMHg)

Due to the complexity of the analytical methods and the extremely low environmental levels, MeHg observations in the marine environment are rare. Additionally, they are the most uncertain observations. Here, to calculate the MQO, we assume an uncertainty of 50%. We evaluate the models predictive capabilities in reproducing (1) MeHg concentrations and (2) the methylated Hg fraction $M_{frac}$ = MeHg/Hg$_T$. The latter allows us to evaluate the modelled net methylation independent of the Hg$_T$ model error (Table 10). In the Baltic Sea MeHg observed concentrations are in the range 191 fM (20 – 603) while the modelled range is 213 fM (57 – 350). For $M_{frac}$ the observed range is 11.4% (1.3 – 30.4) (10$^{th}$ and 90$^{th}$ percentiles) while the model predicts 9.9% (3.6 – 20.2). The frequency distribution for observed and modelled $M_{frac}$ is given in Figure 15. The model is in very good agreement with the observations on average but cannot reproduce the observed extreme values. In total there are 17 (6.5%) samples with a $M_{frac}$ between 33% and 73%, all of which were measured in the intermediate layer between 70 and 160m.

Evaluating the relative $M_{frac}$ metric instead of absolute MeHg concentrations reduces the systematic error from -28% to 5% and the amplitude error from -74% to -55%. This shows, that the Hg$_T$ bias accounts for roughly 80% of the MeHg systematic error and 50% of the amplitude error. Yet, using $M_{frac}$ has no significant effect on the random error indicating a non-linear relationship between the methylated fraction and the absolute amount of MeHg. While systematic error and amplitude error are comparable to the other Hg species, the random error is much larger (NCRMSE = 1.9). This shows that the methylation-demethylation dynamics in the model is too simplified, pointing to missing processes in the model. Figure 14 depicts MeHg and $M_{frac}$ vertical profiles for the central Baltic Sea deep basins in different years and seasons together with oxygen concentrations. Again, it can be seen, that the model is able to reproduce the average vertical profiles but is incapable of capturing the high and low values. Observations indicate an MeHg hotspot near the oxycline. Here, $M_{frac}$ can become as large as 100%, meaning that there almost all mercury is present as MeHg. The highest MeHg observations coincide with anoxic conditions indicating that the availability of dissolved HgS drives methylation in these regions (Soerensen et al., 2018). In Figure 15, anoxic regions are indicated by negative oxygen concentrations. These are based on measurements of H$_2$S and the net oxygen is calculated based on the reaction: $H_2S + 2O_2 \rightarrow H_2SO_4$.

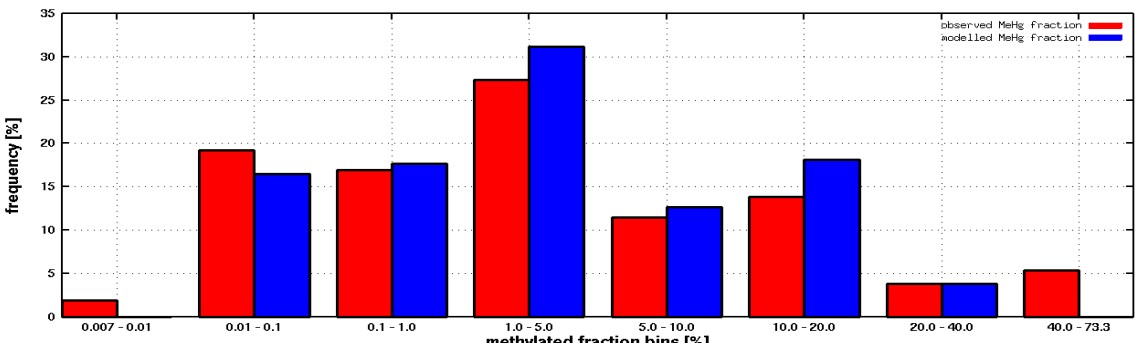

*Figure 14: Observed (Kuss et al., 2017 and Soerensen et al. 2018) and modelled frequency distribution of the methylated Hg fraction $M_{frac}$..*



| Depth [m] | obs MeHg$^+$ | mod MeHg$^+$ | NMB | NCMRSE | NMSD | FAC2 | MQO | obs M$_{frac}$ | mod M$_{frac}$ | N |
|---|---|---|---|---|---|---|---|---|---|---|
| **Total** | **191** | **213** | **0.11** | **1.9** | **- 0.55** | **53 %** | **0.98** | **11.4 %** | **9.9 %** | **160** |
| **March** | | | | | | | | | | |
| **0 – 250** | **222** | **279** | **0.28** | **2.1** | **- 0.55** | **54 %** | **1.03** | **12.0 %** | **10.3 %** | **96** |
| 0 – 50 | 73 | 88 | 0.22 | 1.3 | - 0.54 | 60 % | 1.00 | 6.3 % | 5.2 % | 40 |
| 50 – 150 | 230 | 257 | 0.12 | 1.7 | - 0.67 | 47 % | 0.95 | 15.0 % | 12.2 % | 45 |
| 150 – 250 | 734 | 1067 | 0.45 | 1.5 | - 0.88 | 64 % | 1.09 | 19.8 % | 21.2 % | 10 |
| **Jul – Sep** | | | | | | | | | | |
| **0 – 250** | **163** | **130** | **- 0.20** | **1.6** | **- 0.56** | **49 %** | **0.76** | **10.7 %** | **9.5 %** | **64** |
| 0 – 50 | 53 | 42 | - 0.20 | 1.6 | - 0.67 | 59 % | 0.83 | 6.3 % | 5.0 % | 29 |
| 50 – 250 | 263 | 144 | - 0.45 | 1.4 | - 0.70 | 46 % | 0.73 | 17.8 % | 12.9 % | 28 |
| 150 – 250 | 248 | 174 | 0.11 | 4.2 | - 0.68 | 17 % | 3.21 | 3.80% | 20.4 % | 7 |

*Table 10: Evaluation of seasonally and vertically clustered M$_{frac}$ observations against model values.*

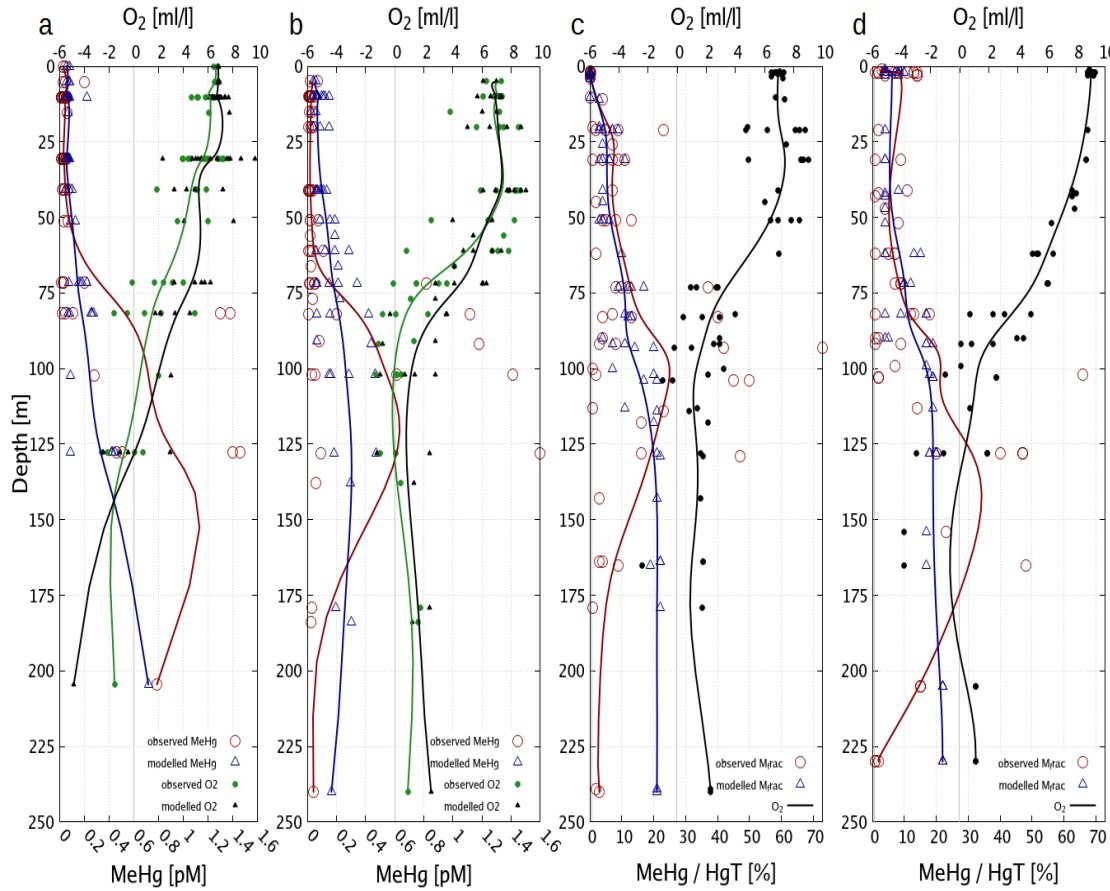

*Figure 15: Vertical MeHg profiles for Baltic deep basins. Negative oxygen concentrations indicate sulfide concentrations. a) MeHg profiles 2014 (Soerensen et al., 2018), b) MeHg profiles 2015 (Soerensen et al., 2018), c) M$_{frac}$ March (Kuss et al., 2017), d) M$_{frac}$ July/August (Kuss et al., 2017).*





The model can reproduce seasonality and vertical gradient of the methylated fraction. On the one hand photolytic demethyla-
tion leads to lower MeHg concentrations in the surface ocean during summer. On the other hand, biological activity leads to

increased MeHg formation in spring and summer. We find, that a biologically induced methylation parameterized with bio-
mass or phytoplankton concentration leads to spring becoming the season with the most effective net methylation. By linking
biological methylation to the remineralization of organic carbon, we introduce a temperature dependency that shifts this to-
wards summer (Figure 16) (Eq. 9, Section 2.3.1). Yet, the model still overestimates methylation in spring and underestimates
methylation in summer. For a more detailed analysis, we look at surface layer MeHg concentrations on four specific days.

Figure 17 depicts MeHg measurements for 21st March and 1st August of the years 2014 and 2015. In March MeHg concen-
trations are between 40 and 300 fM and in August between 10 and 200 fM with pronounced spatial gradients. This 'spotti-
ness' of the MeHg concentrations partially explains the large random error in the model. Moreover, while the general patterns
are similar, methylations shows a significant inter annual variability (Figure 17).

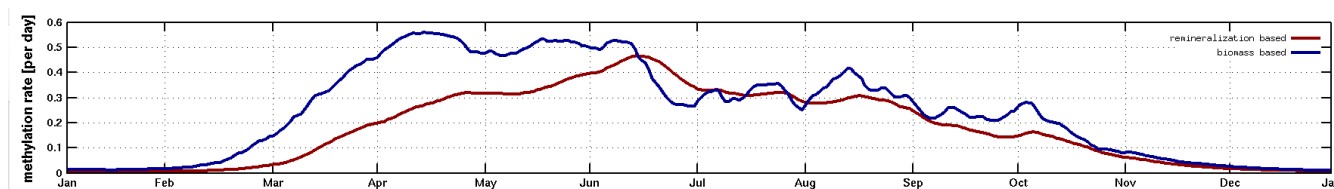

*Figure 16: Seasonality of the biologically induced methylation (R12, Table 2)*

Overall, the model reproduces 53% of MeHg values within a factor of 2. We find that the model performance (MQO = 0.98)
is still within the quality criterion. This is due to the much higher uncertainty of MeHg measurements for which we assumed

an error of 50%. This indicates that further model improvement will be difficult unless more frequent and more precise
MeHg measurements become available. Moreover, to reach their full potential MeHg observations need to be combined with
extensive auxilliary data. This starts with simple parameters like incoming solar radiation to determine the actual intensity of
photolysis or better estimates for special partitioning coefficients for MeHg. In our model, for example, we use a lower $k_d$ of
$MMHg^+$ compared to $Hg^{2+}$ which means that particle settling will increase $M_{frac}$ with increasing depth (Table 4, Section

3.2.3). Moreover, chemical parameters such as $O_2$ and $H_2S$ concentrations have been shown to impact the availability of inor-
ganic $Hg^{2+}$ species for methylation. And finally, microbiological observations ranging from chlorophyll concentration to RNA
showing the activity of methylating bacteria could improve variable methylation rates. From our model evaluation it seems
clear that fixed methylation and demethylation rates cannot account for the observed variability in both MeHg concentrations
and fraction $M_{frac}$. Here, we need a better understanding of the parameters modulating methylation and demethylation rates.




*Figure 17: Methylmercury concentrations in the surface ocean on a) 21st Mar 2014 b) 1st Aug 2014 c) 21st Mar 2015 d) 1st Aug 2015 superimposed are all observations in depths of 0-50m (Table 9).*




### 3.4.5 Hg in biota

Figure 18 depicts annual average Hg loads in the different ecosystem biota species. The North Sea exhibits higher Hg loads in biota which can be explained by the high Hg load from rivers, especially Elbe and Schelde, the lack of permanent sedimentation and the earlier onset and higher overall primary production which increases the effectiveness of the active uptake pathway. The average amount of Hg in biota ranges from 1% to 5% of the $Hg_T$ with higher values in the highly productive

North Sea. During winter only a little Hg is bound in biota due to the low biomass while in summer the fraction can be up to 10%. Due to the high transfer efficiency of $MMHg^+$ (97%), on average, between 5% and 20% of the total $MMHg^+$ is accumulated in biota. In highly productive regions the amount of $MMHg^+$ inside biota can even be larger than the $MMHg^+$ remaining in the water column. Flagellates (Figure 18b) are the most abundant phytoplankton species and thus the most important primary accumulator. However, the diatom bloom occurs earlier in the year and removes MeHg from the water column be-

fore the flagellate bloom. The higher Hg load in diatoms (Fig 18a) is due to their lower carbon content. Finally, cyanobacteria (Figure 18c) which can lead to major blooms in late summer / early autumn are the dominant species later in the year and MeHg loads during the bloom exceeds those of the diatoms and flagellates. Due to the active Hg uptake, micro (Figure 18d) and meso zooplankton (Figure 18e) have a higher accumulation factor than the phytoplankton species. Finally, figure 18f depicts the Hg load in fish.

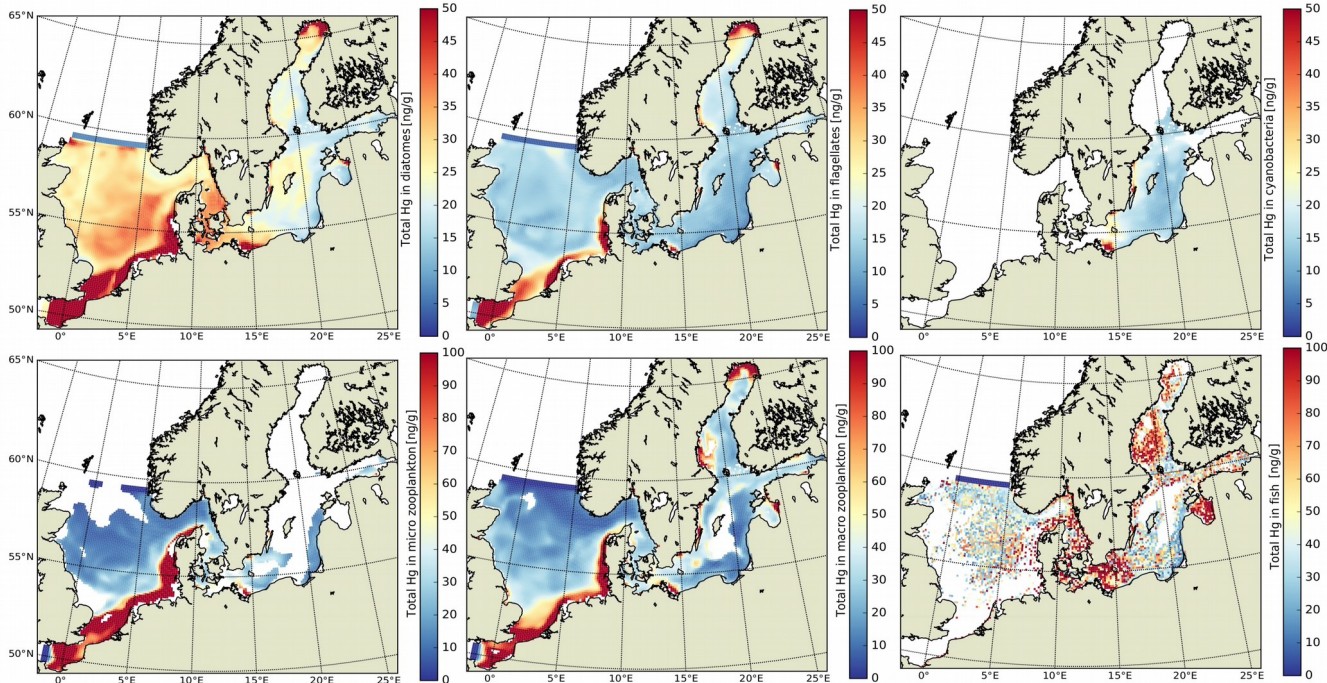

*Figure 18: annual average $Hg_T$ concentrations [ng/l] in biota: a) diatoms, b) flagellates, c) cyanobacteria, d) micro zooplankton, e) meso zooplankton, f) fraction of Hg in fish.*





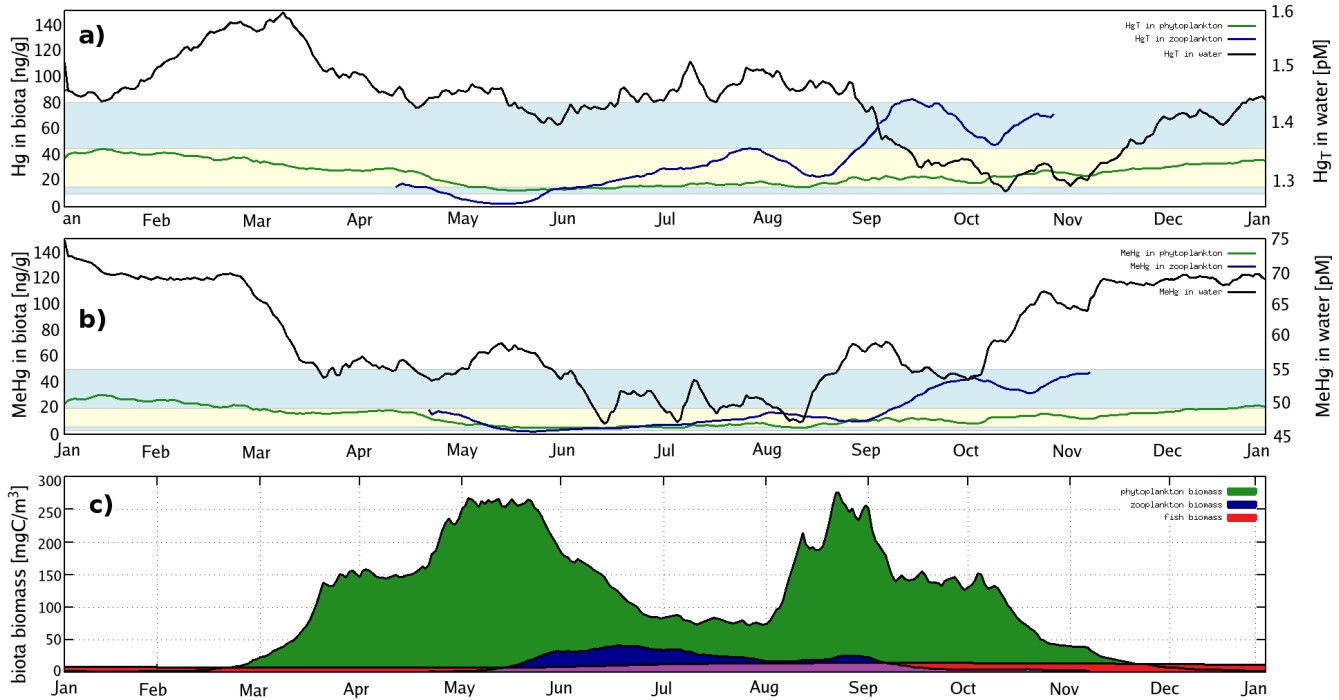

*Figure 19: Seasonality of modelled a) Hg and b) MeHg loads in phytoplankton and zooplankton. Superimposed is the water concentration of a) Hg and b)= MeHg. c) gives the integrated biomass. All values are averages for the Baltic Sea integrated over the top 100m.*

As the last step of the model evaluation, we compare HgT and MeHg loads in biota to observations. Field studies investigating the total Hg content of biota are fairly common and can be used to estimate the model bias. However, only little data on MeHg in biota and species diversity is available. On average the observed HgT content in phytoplankton lies in the range of 0.002±0.001 μg/g wet weight and that for zooplankton in the range of 0.006±0.005 μg/g wet weight (Nfon et al. 2009). Here, we use a conversion factor for wet weight (w.w.) to dry weight (d.w.) of 0.2 for phytoplankton and 0.16 for zooplankton (Cushing 1958; Ricciardi and Bourget 1998). Moreover, biomass in ECOSMO is defined in mgC, whereas observations are reported in mg wet weight (ww) or mg dry weight (dw). We estimate the ratio of mgC to mg dw as: 0.2 for diatoms, 0.33 for flagellates and cyanobacteria, and 0.5 for zooplankton (Sicko-Goad et al. 1984; Walve and Larsson 1999). With this, we estimate the expected average HgT loads in biota in the Baltic Sea in the range of 30 (15–45) ng/gC in phytoplankton and 75 (10–120) ng/gC in zooplankton. MeHg loads in phytoplankton are expected to be around 10 (5-15) ng/g d.w. while being larger for cells with a larger surface to volume ratio (Pickhardt and Fisher 2007; Soerensen et al. 2016). Figure 19 depicts the average HgT and MeHg loads in phytoplankton and zooplankton. For phytoplankton, the model lies well within the expected range for HgT (25-80 ng/gC) (Figure 19a) and MeHg (5-15 ng/gC) (Figure 19b). During winter when phytoplankton biomass is low (Figure 19c), Hg loads reach the maximum of the expected bioaccumulation range and once production starts growth





dilution lowers the modelled HgT and MeHg loads and their concentrations in the water column declines by 10% and 20% respectively. For zooplankton, values are well within the expected range. They start low in the beginning and rise over the year. At the end of the year, cyanobacteria start to dominate the phytoplankton community leading to higher phytoplankton MeHg concentrations and another decline in marine HgT concentrations. At the same time, the decrease in diatom and flagel-

late concentration lead to an increase in the fraction of microzooplankton of the mesozooplankton diet, increasing their trophic level and further increasing the zooplankton HgT load. Finally, with Hg loads between 70 and 140 ng/gC dw, fish are the highest trophic level in the ecosystem model. Due to the much more efficient active uptake of $MMHg^+$ compared to $Hg^{2+}$ in fish 80 to 90% of the accumulated Hg is in the form of $MMHg^+$.

Next, we evaluate the model capabilities to reproduce Hg content in fish. For this, we compare the modelled bioaccumulation

in the functional ecosystem group representing fish to herring. This pelagic species corresponds best to the fish functional group implemented into ECOSMOS (Daewel and Schrum, 2019). The analysis is based on 1166 measurements of Hg in fish muscle tissue. We use the same conversion factors as for zooplankton to convert the model carbon dry weight to wet weight total biomass (1 ng/gC d.w. = 3.125 ng/g ww). For this, the dataset is split into five Baltic Sea regions:

(1) The Swedish West Coast, a stripe from Goteborg to Oslo

(2) The Southern Baltic Proper which includes the Bornholm Sea and the southern Gotland Sea

(3) The Northern Baltic Proper which includes most of the Gotland Sea

(4) The Bothnian Sea

(5) The Bothnian Bay

It is not possible to compare the catched fish to an individual model grid cell and timestep. Therefore, we compare to ob-

served average $Hg_{Fish}$ concentrations. The model reproduces the observed average $Hg_{Fish}$ of 28 ng/g in the Baltic Sea with a systematic error of -9% ($Hg_{Fish}$ = 25 ng/g) (Table 11). In order to estimate the model variability of $Hg_{Fish}$ for each region we vertically integrate annual average model values for each grid column. The result is a fish dataset in which each member rep-resents a model fish that has spent its life in a single 10x10 km² water column. In reality, herring are not confined to 10x10 km² grid cells and their Hg accumulation depends on their migration patterns. Yet, we argue that this approach approximates

the model spread (Figure 20). This allows us to calculate not only the bias but also to estimate the model standard deviation. On average, over the whole Baltic Sea, the model captures the observed variability (NMSD = 9%). The error is driven by the West Coast region (NMSD = 309%) while it varies between (29% – 76%) in the remaining Baltic Sea. In the West Coast re-gion, the observed fish exhibit less than half the variability observed in all other regions. While the model captures the vari-ability in the other regions it shows the opposite behavior as the observations in the West Coast. In this shallow region, we

explain the high model concentrations with regular Hg resuspension from sediments which create pockets of elevated $Hg_T$ and MeHg concentrations. Thus, the large model spread is an artefact of our methodology based on static fish.





*Figure 20: Modelled and observed frequency distribution of Hg in fish in Baltic Sea regions (Soerensen and Faxneld 2020).*

|  | **Baltic Sea** | **West Coast** | **Southern Proper** | **Northern Proper** | **Bothnian Sea** | **Bothnian Bay** |
|---|---|---|---|---|---|---|
| **Obs mean** | **28** | 25 | 22 | 22 | 34 | 39 |
| **Mod mean** | **25** | 28 | 19 | 24 | 27 | 28 |
| **NMB** | **− 9 %** | 12 % | -14 % | 7 % | − 19 % | − 29 % |
| **Obs stdev** | **19** | 9 | 21 | 19 | 22 | 19 |
| **Mod stdev** | **32** | 35 | 36 | 30 | 29 | 24 |
| **NMSD** | **9 %** | 309 % | 67 % | 58 % | 31 % | 29 % |
| **N** | **1166** | 302 | 264 | 178 | 248 | 174 |

*Table 11: Comparison of modelled and observed average Hg concentrations in Baltic Herring (Soerensen and Faxneld*
*2020).*



## 4 Conclusions

In this paper, we present the regional scale 3D high-resolution biogeochemical multi-media Hg model MERCY v2.0. The numerical model combines hydrodynamic models for atmosphere and ocean including a marine ecosystem model. MERCY includes a comprehensive marine Hg scheme to calculate transport, transformation and bio-accumulation. The schemes for

chemistry, partitioning, and bioaccumulation are based on literature values and no domain-specific model tuning has been done. We would like to emphasize that MERCY is applicable for any marine region or even for global application. The major factors when applying the MERCY model to other regions are (1) partitioning coefficients to organic material (OM) as the type of OM varies regionally, (2) the parametrization for biogenic reduction as the values presented here are based on cyanobacteria in the Baltic Sea, and (3) the ecosystem model, as trophic dynamics and phytoplankton uptake rates can vary

widely between regions. To our knowledge, it is the first model capable of linking atmospheric Hg emissions to MeHg accumulation in higher trophic levels. The intention of this initial model publication is the detailed presentation of the model and first results focusing on model performance evaluation and the identification of the processes and parameters responsible for the model error. A more comprehensive analysis of the dynamics and variability of Hg speciation, partitioning, and bioaccumulation is required for future studies. While our model performs more realistically than earlier models for marine Hg cyc-

ling, there are still large uncertainties, especially regarding methylation.

| Hg species | mean ($10^{th}$ – $90^{th}$ percentile) | | model error | | | performance criteria | | |
| --- | --- | --- | --- | --- | --- | --- | --- | --- |
| | observation | model | systematic | random | amplitude | FAC2 | MQO | N |
| $Hg_{Filt.}$ | 2.69 pM (0.9 – 6.0) | 2.24 pM (1.1 – 3.7) | - 17 % | 67 % | - 55 % | 72 % | 1.44 | 435 |
| $Hg^0$ | 73.2 fM (53 – 99) | 74.6 fM (52 – 123) | 2 % | 35 % | 38 % | 97 % | 0.84 | 477 |
| MeHg | 190 fM (20 – 612) | 135 fM (48 – 270) | - 28 % | 160 % | - 74 % | 49 % | 0.80 | 264 |
| MeHg / $Hg_T$ | 11.4 % (1.3 – 30) | 9.9 % (3.6 – 20) | - 5 % | 190 % | - 55 % | 53 % | 0.98 | 160 |
| Hg in fish | 28 ng/g (12 – 52) | 25 ng/g (6 – 71) | 3 % | n/a | 9 % | n/a | n/a | 1166 |

*Table 12: Model performance evaluation for the filtered Hg fraction, dissolved elemental Hg, and the methylated Hg fraction. The model error is separated into systematic error (normalized bias), random error (normalized centred root mean square error), and amplitude error (normalized mean standard deviation). The model quality objective target value is MQO < 1.0 (consult Section 3.1 for more information).*

We evaluated model performance for key Hg species based on a simulation for the North and Baltic Sea for the years 2000 to

2016. We chose these regions due to the availability of observations. Moreover, the two regions covers a range of regimes, has high primary productivity and is relevant for fisheries. Unlike atmospheric Hg modelling, there is no precedent or scientific consensus defining the state-of-the-art requirements and limitations of reproducing concentrations of different marine Hg species. Considering the inherent uncertainty of a comparison of model values and observed concentrations (e.g. measurement error, sampling error, error of the hydrodynamic models, the uncertainty of reaction rates, and unknown processes)





we define model values within a factor of 2 of the observations as a reasonable agreement. Moreover, we used a statistical model quality objective (MQO < 1.0) to assess the model skill (Carnevale et al., 2014) (Section 3.1).

A detailed model performance evaluation for the North and Baltic Sea demonstrates that the model can reproduce concentrations and seasonality of single Hg species to a degree that validates the model predictive capabilities. For $Hg_T$, which we evaluated based on measurements of the filterable Hg fraction the model is able to reproduce 72% of the observations within

a factor of 2 (Table 12). We find that the model can reproduce background concentrations in the open parts of the shelf seas (1.0 – 1.5 pM). The model error can mostly be attributed to random and amplitude error. The main source of uncertainty in the model is the transport dynamics of the large Hg influx from rivers and the Wadden Sea. These lead to observed Hg peaks of up to 10pM. The model resolution of 10x10km proved insufficient to reproduce the observed temporal and spatial gradients. Because the majority of observations are at the coast near major rivers, the model does not reach the quality objective

(MQO = 1.44). Moreover, in the Baltic Sea, the model overestimates vertical mixing from deeper regions with elevated Hg concentrations. This is caused by the coarse vertical resolution below 150 m which leads to numerical diffusion and an underestimation of stratification. We summarize the improvement of the model performance for $Hg_T$ requires optimizing of the hydrodynamic model. Unless circulation patterns, stratification seasonality, resuspension events and upwelling regions are correctly represented hardly any improvement of the model can be achieved. Further, for the coastal ocean, we find that river

inflow needs to be better resolved, ideally with daily loads including fluxes of dissolved and particulate carbon. Finally, particle partitioning and subsequent sedimentation is a major source of uncertainty. We achieved better results using a $\log(k_d)$ of 6.6 (Tesan et al., 2020) which is an order of magnitude higher than those used by other models.

The model performed best for elemental $Hg^0$. Due to air-sea exchange, $Hg^0$ is the key species controlling the exchange between atmosphere and ocean. Any bias in modelled $Hg^0$ fields directly influences the marine total Hg budget and leads to

unrealistic results. MERCY is able to reproduce 97% of $Hg^0$ measurements within a factor of 2. We find that the chemical (often referred to as dark) and photolytic reduction processes produce roughly the same amount of $Hg^0$ annually although with different seasonality. Moreover, elevated $Hg^0$ concentrations in the Baltic Sea between July and October could be reproduced by implementing biological reduction by cyanobacteria. Finally, we find that it is important to consider temperature dependence for the chemical reduction reaction to correctly reproduce the seasonality. With a model skill of MQO = 0.84 we

conclude that the model performance for $Hg^0$ is in a range where further improvements become marginal. Possible improvements are photolytic reaction rates based on actual wavelengths instead of the photolytic active radiation, and a better understanding of biological reducers.

Evaluation of MeHg resulted in the methylated fraction $M_{frac}$ (MeHg/$Hg_T$) for which 55% of model values are within a factor of 2 of observations. The model is able to reproduce the observed mean and seasonality but is unable to capture the observed

maxima resulting in a large random error. Yet, because of the high measurement uncertainty the model still reaches the quality objective (MQO = 0.98) indicating that the observations are limiting model development. We found that in order to pro-





duce realistic MeHg concentrations throughout the year required methylation occurring in oxic waters. Oxic methylation is the primary or sole source (80% – 100%) of MeHg in large parts of the model domain. The anoxic methylation reaction is dominant in anoxic waters (the deep basins of the Baltic Sea). We found that assumptions made in other models linking

methylation to productivity or chlorophyll concentrations pose two problems:Firstly, they lead to regions with zero MeHg in seasons with no primary production and very low MeHg concentrations in the deep anoxic basins. And secondly, they produce a phase error in the seasonality due to an overestimation of MeHg during the spring bloom. In MERCY, we parameterize the biogenic methylation with the amount of remineralized organic matter, which adds a temperature dependence to the process which in turn reduces the impact of the spring bloom. Moreover, various sensitivity runs using varying parameters to

modulate the biogenic methylation rate to test for possible biological drivers have failed to surpass model formulations including a constant oxic methylation reaction. We summarize that poor model performance for MeHg is the key source of uncertainty in the presented model. In order to improve the model performance a more detailed understanding of methylation processes is required. Moreover, more high-quality observations, especially on MeHg seasonality are needed to allow for model-based process studies. The addition of isotopic fractionation to the model might also help to further constrainment of

sources and sinks of MeHg.

Finally, we evaluate the model's ability to reproduce Hg in biota. Our model provides Hg and MeHg loads in phytoplankton, zooplankton, and fish which are inside of the observed range. We find that the modelled phytoplankton concentrations are varying within the observed maximum and minimum loads. Zooplankton changes in trophic level over the course of the year due to changes in diet. As expected, the model predicts the highest MeHg loads in fish making up 90% of the total Hg in fish

due to its high transfer efficiency. Most parameters used for bioaccumulation are highly uncertain and there is ample room for improvement in this part of the model. We hypothesize that the ecosystem model which is focused on correctly reproducing carbon fluxes needs improvements regarding functional traits relevant for bioaccumulation such as size, shape, or feeding behaviour.

The presented model allows hypothesis testing within a consistent physical-biological-biogeochemical framework based on

basic principles. We are currently working on a model version that allows for seamless coupling with different hydrodynamic ocean and marine ecosystem models to increase the applicability of the model. The model performance is here only cursory evaluated to limit the length of the paper For the future, we plan to investigate the sources of model uncertainty and sensitivity in order to identify the unsufficient understanding of the processes and find out the imprecise or unknown parameters, especially concerning methylation and biological uptake. Finally, we want to employ and promote the MERCY model as a tool

for hypothesis testing and prediction within a consistent physical-biological-biogeochemical framework based on basic principles. This will enable researchers to (1) improve our understanding of the natural variability from seasonal to decadal time scales, (2) investigate forcing dynamics, leading to MeHg accumulation in seafood and (3) to estimate the impact of anthropogenic and natural drivers in support of the Minamata Convention on mercury.





## Author contributions

| Contributor role | Role definition *following CrediT taxonomy: https://credit.niso.org/* | Authors |
|---|---|---|
| **Conceptualization** | Ideas; formulation or evolution of overarching research goals and aims. | |
| | … for the mercuy model | JB, CS |
| | … for the bioaccumulation model | DA, UD, JB |
| **Methodology** | Development or design of methodology; creation of models | |
| | … Hg chemical mechanism | JB, JK, ALS |
| | … Hg and MeHg bioaccumulation | DA |
| **Software** | Programming, software development; designing computer programs; implementation of the computer code and supporting algorithms; testing of existing code components. | |
| | … for HAMSOM and ECOSMO | UD |
| | … for MERCY | JB |
| **Validation** | Verification, whether as a part of the activity or separate, of the overall replication/reproducibility of results/experiments and other research outputs. | JB |
| **Formal analysis** | Application of statistical, mathematical, computational, or other formal techniques to analyze or synthesize study data. | |
| | … for $Hg_T$ and MeHg concentrations | JB |
| | … for $Hg^0$ concentrations and air-sea exchange | JB, JK |
| | … for Hg and MeHg bioaccumulation in lower and higher trophic levels | DA, ALS |
| **Investigation** | Conducting a research and investigation process, specifically performing the experiments, or data/evidence collection. | JB, JK, ALS |
| **Resources** | Provision of study materials, reagents, materials, patients, laboratory samples, animals, instrumentation, computing resources, or other analysis tools. | CS |
| | … Hg and MeHg observational data | JK, ALS |
| **Data Curation** | Management activities to annotate (produce metadata), scrub data and maintain research data (including software code, where it is necessary for interpreting the data itself) for initial use and later reuse. | ALS, JK, UD |
| **Writing – original draft preparation** | Creation and/or presentation of the published work, specifically writing the initial draft (including substantive translation). | JB, ALS, DA |
| **Writing – review and editing** | Preparation, creation and/or presentation of the published work by those from the original research group, specifically critical review, commentary or revision – including pre- or post-publication stages. | ALS, CS, DA |
| **Visualization** | Preparation, creation and/or presentation of the published work, specifically visualization/data presentation. | JB |
| **Supervision** | Oversight and leadership responsibility for the research activity planning and execution, including mentorship external to the core team. | JB, CS |
| **Project administration** | Management and coordination responsibility for the research activity planning and execution. | JB, CS |
| **Funding acquisition** | Acquisition of the financial support for the project leading to this publication. | CS, JB |



## Code availability

**Source code:** The MERCY v2.0 source code is available upon request at https://zenodo.org/record/7101217

**DOI:** 10.5281/zenodo.7101217

**Contact:** johannes.bieser@hereon.de

**Licensing:** MERCY v2.0 is dual-licensed under the Apache License, Version 2.0 and the Gnu Public License, Version 3.0

**Driving models:**

*COSMO-CLM* is free of charge forall research applications. (Sørland et al., 2012) Access is license-restricted: http://www.cosmo-model.org/content/consortium/licencing.htm

*CMAQ* is an active open-source development project of the U.S. EPA that consists of a suite of programs for conducting air quality model simulations. The model is freely available at https://cmascenter.org/

*HAMSOM-ECOMSO* model code access and data can be ob-tained upon request (Daewel and Schrum, 2019). The code is available from the Helmholtz Centre Geesthacht Git repository https://coastgit.hzg.de/udaewel/hamsom-ecosmoe2e/

## Competing interests.

The authors declare that they have no conflict of interest.

## Acknowledgements

We want to thank all data providers for their invaluable input without which we would not have been able to develop the MERCY model. Special thanks to Lars-Eric Heimbürger-Boavida for the fruitful discussions on marine Hg cycling and Franz Slemr for sharing his vast knowledge on atmospheric chemistry and Hg cycling.

This work was funded by the H2020 project iGOSP under the ERA-PLANET program (Grant agreement no: 689443) and the Marue Sklodowska-Curie Innovative Training Network GMOS-TRAIN (Grant agreement no: 860497). ALS acknowledge financial support from the Swedish Research Council Formas (grant no: 2021-00942).





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
