# Peer review of "The 3D biogeochemical marine mercury cycling model MERCY v2.0 – linking atmospheric Hg to methyl mercury in fish."

_Geoscientific Model Development, 2021_

## Author Comment (AC1)

**Reviewer #1: https://doi.org/10.5194/gmd-2021-427-RC1**

We want to thank the reviewer for carefully reading our manuscript and for thoroughly checking references, equations, and numbering. Especially given the length of this mansucript. It seems we had an error in the numbering of equations, figures, and tables. We carefully went through the whole manuscript again and to our best knowledge have identified and corrected all typos and mistakes therein. In the following we describe in detail how we addressed the remarks:

**General comment:**

The manuscript presents detailed description of formulation, development and evaluation of a new biogeochemical marine Hg cycling model MERCY v2.0 as a part of a multi-media modelling system. Developments of multi-media capabilities of Hg dispersion modelling is highly topical. The problem of Hg pollution on a global scale is well recognized and currently assessed under the effectiveness evaluation efforts of the Minamata Convention. Despite other pollutants Hg requires model evaluation in various environmental compartments. However, available developments of Hg modelling in the marine environment are still insufficient. The presented a model of Hg cycling in seawater including transport transformation and bioaccumulation processed. The model is applied as a part of a modelling complex in combination with atmospheric and oceanic transport models, and a seawater biogeochemical model to simulate Hg levels and dynamics in the North and Baltic seas. The results are thoroughly evaluated against observations to reveal the model uncertainties and propose ways for further improvement. For this purpose, a system of detailed statistical analysis is developed and applied based on methods used in atmospheric transport modelling. This statistical evaluation system could be useful for application by other marine chemistry modelers.

The subject of the manuscript is relevant to the scope of the journal and the work makes up a new and original contribution to the modelling science. The scientific approaches applied are adequate and explicitly stated. Description of the modelling methods is sufficiently complete and precise to allow reproduction. The manuscript will be suitable for publication after addressing comments mentioned below.

**Specific comments:**

Generally, the manuscript contains a large number of typos and misprints and requires careful editing.

**A: We checked the whole manuscript and corrected all errors to our best knowledge.**

Page 3, lines 84: "While there is a large number of emissions ..."

Probably, there should be mentioned a large number of emission inventories.

**A: Corrected**

Page 7, lines 182: "... change in concentration of Hg state variables over time  $\delta C/\delta t$  is estimated by the prognostic equation..."

 $\delta C/\delta t$  is unnecessary here. The partial derivative describes the change rate. The change itself requires integration of the equation over time.

**A: Thanks for pointing out this inaccuracy. We now make clear that we talk about the rate of change so that the equation is correct.**

Page 9, lines 220-227: "… *Bioconcentration* … *remineralization rate (see Eq. 9 in Section 2.3.1)*. …" Notations of variables and parameters used in this paragraph differ from those in Eq. 5. It complicates understanding.

**A:** We now use the same variable  $k_{rem}$  in both equations for the remineralization rate.

Page 10, Figure 1: The oxidation pathway via formation of the intermediate oxidation product (Hg\*) is not included to the model (page 13, line 280) but shown in the model scheme.

**A: We corrected the figure.**

Page 12, Table 3: Reactions R5, R13, R18 and R20 are not shown in the model scheme (Fig. 1).

**A: We corrected this in figure 1. Please note that the R20 is extending left from Hg0 out of the figure. We clarified this in the caption.**

---

## Author Comment (AC2)

**Dear Yanxu,**     **Reviewer #2: https://doi.org/10.5194/gmd-2021-427-RC2**

**Thank you so much for taking the time to review our paper. Your expertise and knowledge in the field is greatly appreciated. Your insights and feedback will undoubtedly help us improve our work and make it stronger. We're grateful for your time and effort, and we look forward to incorporating your suggestions into our future research. Thanks again for your support and expertise. In the following we describe in detail how we addressed your comments in the revised manuscript.**

The Marine Hg cycle is an important component of its global biogeochemical cycling. Numerical models, especially 3D ones, are useful to reveal the interaction between transport and biogeochemical processes, interpret observations, and test hypotheses. However, the history of the 3D ocean Hg model is less than 10 years and there are only a handful of such models, which limits our ability to conduct multi-model intercomparison and adopt a model ensemble approach. This manuscript includes a detailed description of the processes and configuration of a numerical multi-compartment model for marine Hg cycling, MERCY v2.0. It presents the model evaluation from regional simulations of two shelf seas. The model also includes important and novel updates regarding Hg biogeochemistry, such as the S-Hg chemistry, larger Kd value consistent with field observations, sedimentation/resuspension, and the tentative inclusion of fish in 3D models. Overall, I consider it an important addition and advancement to existing ocean Hg modeling efforts, which merits publication in GMD. Congratulations to the author team.

Some suggestions and questions:

2.1 The model mainly follows a prognostic equation, and the authors describe each sub-term of the transformation term in great detail. Among them the implementation of the sulfur chemistry of mercury is novel, yet an evaluation of the importance of this newly added process seems to be lacking. How does it compare with observations?

**A: The S-chemistry is only activated in anoxic regions. In our application there are anoxic waters in the deep basins of the Baltic Sea. Without S-chemistry the Hg attached to particles sinks to the ocean floor. With S-chemistry included, Hg-S is formed which is considered to be in the form of dissolved nanoparticles without settling velocity. This means that there is an accumulation of Hg in the anoxic deep waters when using S-chemistry while there is more pronounced sedimentation without it. Comparison with observtions shows that runs without S-chemistry underestimate Hg concentrations in the deep Baltic basins.**

**We added a paragraph at the end of Section 3.4.2:**

*L853-861: „Finally, as the deep basins of the Baltic Sea are anoxic, in this area sulfur chemistry becomes relevant (R6-R9 Table 3). The effect of adding HgS and HgS-DOM to the chemistry scheme leads to particulate Hg-POM transforming into dissolved HgS species. The effect of this is two fold: (1) Firstly, Hg that is scavanged from the stratified surface layer by detritus (biological pump) accumulates directly at the boundary between oxic and anoxic waters.(2) Secondly, as eventually all inorganic Hg is transformed into HgS species, pareticle settling stops being a sink and Hg persists in the water column. Whereas Hg is effectively transported to the sediment in model runs without sulfur chemsitry. This leads to Hg concentrations being constant in the anoxic layer with higher values found only directly at the sea floor. Comparing to observations, we find that the model with sulfur chemistry is better able to capture the observed Hg distribution (Soerensen et al., 2018).“*

**We also added this finding to the conclusions section:**

*L1103-1106: "We found that including sulfur chemistry improves model performance in the deep anoxic water layer in the Batlic Sea basins. The mechanism is, that Hg transported downwards from the stratified oxic und productive surface layer through the biological pump transforms into dissolved HgS species in anoxic waters. This stops the downward gradient and lessens the role of the sediments in this region as a sink."*

2.2 The simulation of the bioconcentration process considers the biological uptake of Hg by organisms from higher trophic levels through the body parts exposed to seawater other than phytoplankton. Can you quantify the contribution from this pathway and via food consumption?

**A: The relative contribution of the two pathways for Hg and MMHg uptake varies over space and time. In our model phytoplankton bioconcentration by definition is 100%. For the other species bioconcentration is responsible for 10% to 20% of Hg uptake and biomagnification for 80% to 90%. We are actually just about to submit a manuscript with a detailed analysis on Hg and MMHg bioaccumulation based on the MERCY model.**

**We now mention this briefly in the mansucript but would refer to our upcoming paper for a detailed answer.**

*L1033-1035: „Looking at the two uptake pathways of bioconcentration and biomagnification we find that biomagnification is responsible for 80 to 90% of the total Hg uptake for non-phytoplankton species. A more detailed analysis can be found in Amptmeijer et al. (2023)."*

*Weo added a reference to our upcoming paper so that it can be linked in the future. We are also happy to share our manuscript with the reviewer in advance.*

*„Amptmeijer, D.J., Mikheeva, E., Daewel, U., Bieser, J., Schrum, C.: The impact of ecosystem interactions on marine mercury and methylmercury concentrations in the North- and Baltic Seas. (in prep.)."*

2.3.1 Although the model is claimed to be improved by employing a high Kd, the authors do not seem to have explicitly considered the effect of the biological pump on the mercury species at different depths.

2.3.2 A single sinking velocity wd is utilized to calculate the vertical transport but the association between this and the biological pump was not given in detail.

2.3.3 Nevertheless, including the sedimentation and resuspension in the model makes it more complete than previous models.

**A: We agree that this can be formulated more clearly. Basically, the $k_d$ value in our model can be seen as the efficiency factor of the biological pump. We did test the model without the process by turning the biological pump off (i.e. $k_d$ lim $\rightarrow$ 0) and without this process the observed Hg depletion in Baltic surface waters cannot be reproduced at all.**

**About the suggested analysis of different depths: In this region there are but two water masses with relatively constant Hg concentrations. The surface layer has generally low Hg concentrations between 0.5 pM in winter and 1.5 pM in summer. The lower values in summer are due to the downward transport through the biological pump. The higher concentrations in**

winter are a mixture of mostly atmosperic input but also upwelling of deeper waters in certain regions like the Swedish coast. The lower, anoxic water mass has generally higher concentrations in the range of 2 to 4 pM with occationally higher values near the sediments. This is depicted in Figure 10.

[Figure]

*Figure 10: Vertical seasonality (daily average Hg concentration) profiles in the Baltic Sea.*

Outside the major estuaries and directly at the coastline where a fraction of the Hg from rivers sediments quickly with the terrestrial particles, POC is mostly of biologcial origin (detritus). We think that terrestrial POC might be the reason for the model bias in the Bothnian Bay and will include more detailed terrestrial POC fluxes and interactions in future updates of our model. However, we did perform sensitivity tests on this.

We acknowledge that for a global application the MERCY model should implement a more sophisticated particle settling scheme like a depth dependent sinking velocity and particle ageing. But we argue that in the shallow coastal ocean this does not play a role. Firstly the North Sea depth is <50m in the southern part and <100m in the Northern part. (With the exception of the Norwegian trench that can be up to 400m deep). In the Baltic Sea the deep basins are between 200m and 400m and the remaining part is also mostly <50m (see Figure 6b in the manuscript). As described above, the Baltic Sea is mainly stratified with a surface layer around 50m to a maximum of 100m.

*L809-812: „The two processes governing this are: (1) Stratification and particle settling in the central Baltic deep basins after the onset of primary production. This is the biological pump as POC particles here are mainly of biological origin (detritus). And (2) increased photoreduction and subsequent atmospheric exchange of Hg$^0$ (air-sea exchange)."*

2.4 The quality criteria proposed in this manuscript entails sophisticated statistical analyses, and the elaborated presentation enables other ocean modelers to reproduce and apply. Also, the authors emphasize the importance of observational data and indicate that some processes are poorly constrained in the discussion. This can help field and laboratory studies address these issues.

**A: Thank you for your kind words.**

Specific points (some may also be spotted by other reviewers):

1. Line 81, the authors state "The only real sink for Hg in the environment is a burial in the lithosphere mainly as stable cinnabar (HgS) in anoxic marine sediments." However, there are several data suggesting that the sedimentation of compounds to organic material is a major sink in coastal and open-ocean systems. This may need the authors to include some references to address.

2. Line 106, "red-dox chemistry" should be "red-ox chemistry".
   **A: Corrected**

3. Line 115 and line 116, Rosati 2022 paper was mentioned twice but they did not appear in the reference list.
   **A: Added Rosati et al. 2022 to the reference list.**

4. Line 162, "en-to-end" should be "end-to-end"
   **A: Corrected**

5. Line 171 Table 1, "GOM", "PBM" - it would be better if these abbreviations be written out in full on first use.
   **A: Corrected**

6. Line 270, "concentrationdependent" should be "concentration dependent". Line 271, "raction" should be "reaction". And it would be better to add the note on R12 about the remineralized organic matter concentration dependency.
   **A: Corrected**

7. Line 295, is there a literature-based argument to support the use of negative oxygen concentration to represent sulfur ions concentration?
   **A: This is a neat trick to reduce the number of state variables in the model that consitently leads to questions as it seems strange or even plainly wrong at first glance. The central idea is that reduced sulfur ($S^{2-}$) concentrations are zero if oxygen concentrations are above 0 and vice versa. Thus, a single variable can be used to store them. To distinguish whether the variable represents $O_2$ or $S^{2-}$ the unused first bit, which represents the sign of the variable, is used.**
   **In summary we can replace an 8 byte (64 bit) double variable by utilizing the a single unused bit in the $O_2$ variable. It might not seem like much, but as you know things add up and the Hg model alone already adds 40 state variables to the physical ocean model.**

***From Neumann et al.:*** *„In the model, the oxygen demand and production of oxygen is coupled to nitrogen conversion. The oxygen concentration controls the recycling of dead organic matter (detritus). If the oxygen is depleted, then the nitrate is used to oxidize detritus and, if nitrate disappears, sulphate is reduced to hydrogen sulphide. Hydrogen sulphide is accounted for by negative oxygen equivalents. Reduction of nitrate (denitrification) is counted as a loss of nitrogen in the model.„*

> ***We added the source to the text the references:***
> ***Neumann, T.: Towards a 3D-ecosystem model of the Baltic Sea.***
> ***Journal of Marine Systems 25, 405-419, 2000.***

8. Line 306, "chemistrcy" should be "chemistry".
   **A: Corrected**

9. Line 320, the square symbol in the formula should be the superscript of "T_w" instead of "w".
   **A: Corrected**

10. Line 348, what does "R(C,B)" mean here? According to the previous introduction (line 188), it represents the transformation term in the prognostic equation?
    **A: There seems to have been a format problem when storing the formulas in the doc file which we did not notice. We corrected this. The correct terms are the partitioning coefficients is $\log(k_d)$ and $\log(k_l)$ repsectively.**

11. Line 353-359, 392-399 and 463, 559-563, the units (even unitless or 1) are needed to be written out in the description of the variables.
    **A: We now use the notation [1] suggested by reviewer #1 for unitless variables.**

12. Line 420 Figure 2, the labels of the color bar may have gone wrong. '60' appears twice for different colors.
    **A: Corrected**

13. Line 436, "due to the comparably low surface areas of these species", do the authors mean diffusive uptake by zooplankton is less important due to the low surface-to-volume ratio of zooplankton? Since zooplankton generally has a larger diameter, thus larger surface areas but a lower surface-to-volume ratio.
    **A: Yes, we are indeed referring to the surface to volume ration rather than the absolute surface area. The surface area of zooplankton is generally larger than that of phytoplankton. But because they are larger their surface to volumne ratio is lower and thus passive uptake results in a lower Hg burden per gram bodymass then it would for phytoplankton.**

14. Line 455, readers may wish to know about the exact feeding relationship of "17 × Feeding rates for biological species (x) on species (y)" from Table 1, however, the predation related to fish is not mentioned here.

**A: The fish species feeds on meso-zooplankton and macrobenthos. We now clarify this in the text. For more details on the ecosytem model we point the reader to the source as these are part of the ECOSMO model (Daewel et al., 2019). The model is freely available (see ‚*Code Availability*' section).**

*L461-462: „Fish feeds on meso zooplankton and macrobenthos following Daewel et al. (2019)"*

15. Line 628 Eq. 44, what is "k1" here?

    **A: There was some error when exporting the equations. We corrected this and now state that to calculate the fraction of model values within a factor of 2**

    $$\text{FAC} 2 = \frac{1}{N} \sum_{i=1}^{N} n_i$$ **of the observations, the value for each prediction/observation pair**

    $n_i$ **is** $n_i = 1$ **for** $0.5 < \dfrac{prediction}{observation} < 2$ **and otherwise** $n_i = 0$ **.**

16. Line 626 and 640, it seems that the stated "the measurement error to range from 20% (Hg0 and HgT) to 50% (MeHg)" is the same value as "U = measurement uncertainty" that is used to calculate MQO? It would be easier for the reader to understand if it could be phrased consistently.

    **A: We clarified that we only use the measurement uncertainty U for our statistical analysis. After revieweing this issue we decided not to speculate about the sampling error as we do not have sufficient information to estimate it.**

17. Figure 12: the title of the x-axis is missing.

    **A: Corrected**

18. Figure 13: according to line 918, the lower panel should be the profile of the North Sea, but it is not listed in the caption.

    **A: Corrected**

19. Figure 16: the caption and axes are not clear.

    **A: Corrected**

---

## Author Comment (AC3)

Comments by the Editor: https://doi.org/10.5194/gmd-2021-427-CEC1

Dear authors,                                             Juan A. Añel, Geosci. Model Dev. Exec. Editor

Unfortunately, after checking your manuscript, it has come to our attention that it does not comply with our "Code and Data Policy".

https://www.geoscientific-model-development.net/policies/code_and_data_policy.html

There are many problems with the code included in your manuscript.

3.1) First, in your manuscript, you state, "The MERCY v2.0 source code is available upon request"; however, the code is available openly on Zenodo.org. This is good news, but please, you should modify the statement in the text and remove the "upon request".

**A: We corrected this**

3.2) Also, you mention in the README file that the code is released under the Apache License. Notwithstanding, the repository is under the CC-4.0 license. You should modify the license of the repository and include a copy of the Apache License with your code.

**A: We requested a change at zenodo.org and the uploaded readme file now includes the correct license as well as a copy of the license statement. MERCY v2.0 is now published under the CC-4.0 license. The „Code Availability" section has been updates accordingly.**

3.3) About the COSMO-CLM and CMAQ codes: We need that they are archived in a suitable repository too, and the webpages that you mention are not enough. You could want to check if they are already stored in one. For example, there are versions of both of them on Zenodo.org, although I do not know if they are exactly the ones that you have used.

**A: We were lucky and the models (more precisely the model versions) we used are available on zenodo:**

**CMAQ 4.7.1 https://doi.org/10.5281/zenodo.1079879**

**COSMO-CLM 4.0 https://doi.org/10.5281/zenodo.5939757**

3.4) The same applies to HAMSOM-ECOMSO. However, here the problem is worse, as the link that you provided in the manuscript does not work. Therefore, you must provide a suitable link.

Therefore, please, publish the mentioned codes in one of the appropriate repositories, and reply to this comment with the relevant information (link and DOI) as soon as possible, as it should be available for the Discussions stage.

**A: We uplaoded the  HAMSOM_ECOSMO_E2E v1.0 model source code to zenodo:**

   **https://doi.org/10.5281/zenodo.7587005**

3.5) In this way, you must include in a potentially reviewed version of your manuscript the modified 'Code and Data Availability' section and DOIs for the codes, which must be available and open without the need to request access to anyone.

**A: We adjusted the ,*Code and Data Availability'* section. It now includes links and DOIs for:**

      **- CMAQ v4.7.1**              **https://doi.org/10.5281/zenodo.1079879**

      **- ECOSMO-HAMSOM v1.0**    **https://doi.org/10.5281/zenodo.7587005**

      **- COSMO-CLM v4.0**        **https://doi.org/10.5281/zenodo.5939757**

      **- MERCY v2.0**             **https://zenodo.org/record/7101217**